# Optimal Scheduling in General Multi-Queue System by Combining Simulation and Neural Network Techniques

**DOI:** 10.3390/s23125479

**Published:** 2023-06-10

**Authors:** Dmitry Efrosinin, Vladimir Vishnevsky, Natalia Stepanova

**Affiliations:** 1Institute for Stochastics, Johannes Kepler University Linz, 4040 Linz, Austria; 2Department of Information Sciences, Peoples’ Friendship University of Russia (RUDN University), Moscow 117198, Russia; 3V.A. Trapeznikov Institute of Control Sciences of Russian Academy of Sciences, Moscow 117997, Russia; vishn@inbox.ru; 4Scientific and Production Company “INSET”, Moscow 129085, Russia; natalia0410@rambler.ru

**Keywords:** optimal scheduling, heterogeneous queues, Markov decision problem, queue simulation, simulated annealing, neural network

## Abstract

The problem of optimal scheduling in a system with parallel queues and a single server has been extensively studied in queueing theory. However, such systems have mostly been analysed by assuming homogeneous attributes of arrival and service processes, or Markov queueing models were usually assumed in heterogeneous cases. The calculation of the optimal scheduling policy in such a queueing system with switching costs and arbitrary inter-arrival and service time distributions is not a trivial task. In this paper, we propose to combine simulation and neural network techniques to solve this problem. The scheduling in this system is performed by means of a neural network informing the controller at a service completion epoch on a queue index which has to be serviced next. We adapt the simulated annealing algorithm to optimize the weights and the biases of the multi-layer neural network initially trained on some arbitrary heuristic control policy with the aim to minimize the average cost function which in turn can be calculated only via simulation. To verify the quality of the obtained optimal solutions, the optimal scheduling policy was calculated by solving a Markov decision problem formulated for the corresponding Markovian counterpart. The results of numerical analysis show the effectiveness of this approach to find the optimal deterministic control policy for the routing, scheduling or resource allocation in general queueing systems. Moreover, a comparison of the results obtained for different distributions illustrates statistical insensitivity of the optimal scheduling policy to the shape of inter-arrival and service time distributions for the same first moments.

## 1. Introduction

Machine learning algorithms have been used over the last ten years in almost all fields where problems associated with data classification, pattern recognition, non-linear regression, etc., have to be solved. The application of such algorithms has also intensified in the field of queueing theory. While the first steps in the successful application of machine learning to evaluate the performance characteristics of simple and complex queueing systems have already been taken, the total number of works on this topic still remains modest. As for reviews, we can only refer to a recent paper by Vishnevsky and Gorbunova [1] which proposes a systematic introduction to the use of machine learning in the study of queueing systems and networks. Before we formulate our specific problem we would like also to make a small contribution to the popularisation of machine learning in the queueing theory by describing briefly the latest works. In Stintzing and Norrman [2], an artificial neural network was used for predicting the number of busy servers in the M/M/s queueing system. The papers of Nii et al. [3] and Sherzer et al. [4] have answered positively the question regarding whether the machines could be useful for solving the problems in general queueing systems. They employed a neural network approach to estimate the mean performance measures of the multi-server queues GI/G/s based on the first two moments of the inter-arrival and service time distributions. A machine learning approach was used in the work of Kyritsis and Deriaz [5] to predict the waiting time in queueing scenarios. The combination of a simulation and machine learning techniques for assessing the performance characteristics was illustrated in Vishnevsky et al. [6] on a queueing system MMAP/PH/M/N with *K* priority classes. Markovian queues were simulated using artificial neural networks in Sivakami et al. [7]. Neural networks were used also in research by Efrosinin and Stepanova [8] to estimate the optimal threshold policy in a heterogeneous M/M/K queueing system. The combination of the Markov decision problem and neural networks for the heterogeneous queueing model with process sharing was studied by Efrosinin et al. [9]. The performance parameters of the closed queueing network by means of a neural network were evaluated in Gorbunova and Vishnevsky [10]. In addition to the presented results of using neural networks in hypothetical queueing theory models, academic studies in this area with real-world applications have gradually been proposed. For example, the problem regarding the choice of an optimum charging–discharging schedule for electric vehicles with the usage of a neural network is proposed by Aljafari et al. [11]. The main conclusion to be drawn from the previous results obtained via the application of machine learning to models of the queueing theory is that the neural networks cannot be treated as a replacement for classical methods in system performance analysis, but rather as a complement to the capabilities of such an analysis.

The systems with parallel queues and one server are known also as polling systems which have found wide application in various fields such as computer networks, telecommunications systems, control in manufacturing and road traffic. For analytic and numerical results in various types of polling systems with applications to broadband wireless Wi-Fi and Wi-MAX networks, we refer interested readers to the textbook by Vishnevsky and Semenova [12] and the references therein. The same authors in [13] developed their research on polling systems to systems with correlated arrival flows such as MAP, BMAP, and the group Poisson arrivals. In Vishnevskiy et al. [14], it was shown that the results obtained by a neural network are close enough to the results of analytical or simulation calculations for the M/M/1 and MAP/M/1-type polling systems with cyclic polling. Markovian versions of a single-server model with parallel queues have been investigated by a number of authors. The two-queue homogeneous model with equal service rates and holding costs was studied in Horfi and Ross [15], where it was shown that the queues must be serviced exhaustively according to the optimal policy. In research by Liu et al. [16], it was shown that the scheduling policy that routes the server with respect to the LQF (Longest Queue First) policy is optimal when all queue lengths are known and that the cyclic scheduling policy is optimal in cases where the only information available is the previous decisions. The systems with multiple heterogeneous queues in different settings, also known as asymmetric polling systems, have been studied intensively in cases where there are no switching costs by Buyukkoc et al. [17], Cox and Smith [18], where the optimality of the static cμ-rule was proved. This policy schedules a server first to the queue *i* with a maximum weight $c_{i}\mu_{i}$ consisting of the holding cost and service rate. In Koole [19], the problem of optimal control in a two-queue system was analysed by means of the continuous-time Markov decision process and dynamic programming approach. The author found numerically that the optimal policy which minimizes the average cost per unit of time can be quite complex if there are both holding and switching costs. The threshold-based policy for such a queueing system was applied by Avram and Gómez-Corral [20], where the expressions for the long-run expected average cost of holding units and switching actions of the server were given. The queueing system with general service times and set-up costs which have an effect on the instantaneous switch from one queue to another was studied in Duenyas and Van Oyen [21]. The authors proposed a simple heuristic scheduling policy for the system with multiple queues. A rather similar model is described in Matsumoto [22], where the optimal scheduling problem is solved in a system with arbitrary time distributions. Here, instead of switching costs, the corresponding set-up time intervals required for switching are used. The system is controlled by the Learning Vector Quantization (LVQ) network, see Kohonen [23] for details, which classifies the system state by the closest codebook vector of a certain class in terms of the Euclidean metric. The problem with this approach is the large number of parameters associated with the codebook vectors, where it is normally required that several vectors per class must be estimated for a given control policy using computationally expensive recurrent algorithms.

This paper proposes a fairly universal method for solving the problem of optimal dynamic scheduling or allocation in queueing systems of the general type, i.e., where the times between events are arbitrarily distributed, and in queueing systems with correlated inter-arrival and service times. Furthermore, it can provide a performance analysis of complex controlled systems described by multidimensional random processes, for which finding analytical, approximate or heuristic solutions is a difficult task. The main idea of the paper is to use a multi-layer neural network for server scheduling. The parameters of this neural network trained first on some arbitrary control policy are optimized then with the aim to minimize a specified average cost function. Moreover, such a cost function for systems with arbitrary inter-arrival and service time distributions can only be computed via simulation. We consider this approach, which combines neural networks with simulation technique, to be quite universal to obtain an optimal deterministic control policy in complicated queueing systems. The method is exemplified by some version of a single-server system with parallel queues equipped with a controller for scheduling a server. The system under study is assumed to have heterogeneous arrival and service attributes, i.e., unequal arrival and service rates, as well as holding and switching costs. Systems with arbitrary distributions and switching costs have not yet been considered by other authors. It is assumed in our model that the queue currently being served by the server is serviced exhaustively. The next queue to be served by the server is selected according to a dynamic scheduling policy based on the queue state information, i.e., on the number of customers waiting in each of parallel queues. It is expected that the changing of the serviced queue involves the switching costs. The holding of a customer in the system is also linked to the corresponding cost. Clearly, even with some fixed scheduling control policy, calculating any characteristics of the proposed queueing system with arbitrary inter-arrival and service time distributions in explicit form is not a trivial task. It is also difficult to fix the dynamic control policy defining the scheduling in large systems in a standard way, e.g., through a control matrix that would contain the corresponding control action for all possible states of the system. Therefore, in such a case we consider it justified to solve the problem of finding the optimal scheduling policy with the aim to minimize the average cost per unit of time by combining the simulation as a tool to calculate the performance characteristics of the system with a machine learning technique, where the neural network will be responsible for dynamic control. By training a neural network for some initial control policy, we obtain characteristics of the network in the form of a matrix of weights and a vector of biases. The process of solving the optimal scheduling problem is then reduced to a discrete parametric optimization. The parameters of the neural network must be optimized in such a way that this network can guarantee the minimal values of the average cost functional by generating control actions at decision epochs. For this purpose, we have chosen one of the random search methods, such as simulated annealing, see, e.g., in Aarts and Korst [24], Ahmed [25]. It is a heuristic method based on a concept of heating and controlled cooling in metallurgy and is normally used for global optimization problems in a large search space without any assumption on the form of the objective function. This algorithm was implemented by Gallo and Capozzi [26] specifically for the probabilistic scheduling problem. The algorithm will be adapted for a non-explicitly defined parametric function with a large number of variables defined on a discrete domain.

To verify the quality of the calculated optimal parameters of the neural network, the values of the average cost functional for the markovian version of the queueing system are compared with the results obtained by solving the Markov decision problem (MDP). The general theory on MDP models is discussed in Puterman [27] and Tijms [28]. The details on application of MDP to controlled queueing systems with heterogeneous servers can be found in Efrosinin [29]. The optimal control policy and the corresponding objective function are calculated in the paper via a policy-iteration algorithm proposed in Howard [30] for an arbitrary finite-state Markov decision process. According to the MDP, the router in our system has to find an optimal control action in the state visited at a decision epoch with the aim to minimize the long-run average cost. Note that for our queueing model under general assumptions the semi-Markov decision problem (SMDP) can be formulated. The SMDP is a more powerful model than the MDP since the time spent by the system in each state before a transition is taken into account by calculating the objective function. The objective function must be calculated here also by means of a simulation. In this case, the reinforcement learning algorithm, e.g., *Q*-*P*-Learning, can be applied. The main problem of this approach consists of the fact that many pairs of state and action can remain non-observable for deterministic control policy and as a result the control actions in such states can not be optimized. However, in our opinion, neural networks can also be used to solve this problem which presents a potential task for further research. The SMDP topic is outside the scope of this article but we refer readers to work by Gosavi [31], where one can find a very interesting overview on reinforcement learning and a well-designed classification of simulated-based optimization algorithms.

Summarising our research in this paper we can highlight the following main contributions: (a) We propose a new controlled single-server system with parallel queues where the router uses a trained multi-level neural network to perform a scheduling control: (b) A simulated annealing method is adapted to optimize the weights and biases of the neural network with the aim to minimize the average cost function which can be calculated only via simulation; (c) The quality of the resulting optimal scheduling policy is verified solving a Markov decision problem for the Markovian analog of the queueing system; (d) We provide detailed numerical analysis of the optimal scheduling policy and discuss its sensitivity to the shape of the inter-arrival and service time distributions; (e) The distinctive feature of our paper is the presence of algorithms employed in the form of pseudocodes with detailed descriptions of relevant steps.

The rest of the paper is organized as follows. Section 2 presents a formal description of the queueing system and optimization problem. Section 3 describes the Markov decision problem and the policy-iteration algorithm used to calculate optimal scheduling policy. In Section 4, the event-based simulation procedure of the proposed queueing system is discussed. The neural network architecture, parametrization and training algorithm are summarized in Section 5. Section 6 presents simulated annealing optimization algorithm. The numerical analysis is shown in Section 7 and concluding remarks are provided in Section 8.

The following notations are introduced for use in sequel. Let $e_{j}$ denote the vector of appropriate dimension with 1 in the *j*th position beginning from 0th and 0 elsewhere, $1_{\left\{A\right\}}$ denote the indicator function which takes the value 1 if the event *A* occurs and 0 otherwise. The notations $\min_{i}\left\{a_{i}\right\}$ and $\max_{i}\left\{a_{i}\right\}$ mean the minimum and maximum of the values that *a* can assume, and $\underset{i}{arg min}\left\{a_{i}\right\}$, $\underset{i}{arg max}\left\{a_{i}\right\}$ denote the element index associated, respectively, with the minimum and maximum value.

## 2. Single-Server System with Parallel Queues

Consider a single-server system with *N* parallel heterogeneous queues of the type GI/G/1 and router for scheduling the server across the queues. Heterogeneity here refers to unequal distributions associated with inter-arrival and service times of customers in different queues, as well as unequal holding and switching costs. The queue that is currently being serviced is exhaustively serviced. Denote I={1,2,…,N} as a queue index set. The proposed queueing system is shown schematically in Figure 1.

Denote $\tau_{n,i}$, n≥1 as the time instants of arrivals to queue *i* and $\nu_{i}:=\nu_{n,i}=\tau_{n,i}-\tau_{n-1,i}$, n≥1 as the sequence of mutually independent and identically distributed inter-arrival times with a CDF $A_{i}\left(t\right)$, i∈I. Further denote by $\zeta_{i}:=\zeta_{n,i}$, n≥1, the service time of the *n*th customer in the *i*th queue. These random variables are also assumed to be mutually independent and generally distributed with CDF $B_{i}\left(t\right)$, i∈I. We assume that the random variables $\nu_{i}$ and $\zeta_{i}$ have at least two first finite moments
$$\begin{array}{cc} \; & a_{k,i}=k\int_{0}^{\infty }x^{k-1}\left(1-A_{i}\left(t\right)\right)dt,\;b_{k,i}=k\int_{0}^{\infty }x^{k-1}\left(1-B_{i}\left(t\right)\right)dt,\;k=1,2. \end{array}$$
The squared coefficients of variation are defined then, respectively, as
$$\begin{array}{c} CV_{\nu_{i}}^{2}=\frac{a_{2,i}}{a_{1,i}^{2}}-1,\;CV_{\zeta_{i}}^{2}=\frac{b_{2,i}}{b_{1,i}^{2}}-1. \end{array}$$
This characteristic will be required to provide a comparison analysis of the optimal scheduling policy for different types of inter-arrival and service time distributions. From now it is assumed that the ergodicity condition is fulfilled, i.e., the traffic load $\rho =\sum_{i=1}^{N}\rho_{i}=\sum_{i=1}^{N}\frac{b_{1,i}}{a_{1,i}}<1$.

Let D(t) indicate the sequence number of the queue currently being serviced by the server at time *t*, and $Q_{i}\left(t\right)$ denote the number of customers in the *i*th queue at time *t*, where i∈I. The states of the system at time *t* are then given by a multidimensional random process
(1)$$\begin{array}{c} {\left\{X\left(t\right)\right\}}_{t\geq 0}={\left\{D\left(t\right),Q_{1}\left(t\right),\ldots ,Q_{N}\left(t\right)\right\}}_{t\geq 0} \end{array}$$
with a state space
(2)$$\begin{array}{c} E=\{x=\left(d,q_{1},\ldots ,q_{N}\right):d\in I,\;q_{i}\in N_{0},\;i\in I\}. \end{array}$$
Further in this section, the notations d(x) and $q_{i}\left(x\right)$ will be used to identify the corresponding components of the vector state x∈E. The cost structure consists of the holding cost $c_{i}$ per unit of time the customer spends in queue *i* and the switching cost $c_{i,j}$ to switch the server from queue *i* to queue *j*.

It is assumed that the system states X(t) are constantly monitored by the router which defines the queue index to be serviced next after a current queue becomes empty. In initial state, when the total system is empty, a server is randomly scheduled to some queue. If the *i*th queue to be served becomes empty, such a moment we call a decision epoch, the router makes a decision by means of the trained neural network whether it must leave the server at the current queue or dispatch it to another queue. The routing to an idle queue is also possible. We remind that the server allocated by the router to a certain queue serves it exhaustively, i.e., it is only possible to change the queue if it becomes empty. Denote by A=I an action space with elements a∈A, where *a* indicates the queue index to be served next after the current queue has been emptied. The subsets A(x) of control actions in state $x\in \hat{E}\subset E$ with
$$\hat{E}=\left\{x=E:q_{d}\left(x\right)=0\right\}$$
coincide with the action space *A*. In all other states *x* from $E\setminus \hat{E}$ the subsets A(x)={0} includes only a fictitious control action 0 which has no influence on the system’s behavior.

The router can operate according to some heuristic control policies. It could be for example a Longest Queue First (LQF) policy which is a dynamic one and it prescribes at decision epochs to serve the next queue with the highest number of customers. If there are more than one queue with the same maximal number of customers, the queue number is selected randomly. Alternatively, the static cμ-rule, which needs only the information if a certain queue in non-empty, can be used for scheduling. According to this control policy the queue *i* with the highest factor $c_{i}\mu_{i}$ which is the product of the holding cost and the service intensity, must be serviced next. In the system with totally symmetric queues the former policy is according to  [16] optimal. The latter control policy is optimal due to [17] if there is no switching costs, i.e., $c_{i,j}=0$. Otherwise, in case of positive switching costs and asymmetric or heterogeneous queues such policies are not optimal with respect to minimization of the average cost per unit of time.

The main idea of an optimal scheduling in our general model is as follows. We will equip the router with a trained neural network which will inform it on the index number of the next queue to which the server should be routed with the aim to reach formulated optimization aims. Obviously, we can only train the neural network on available data sets, i.e., on some heuristic control policy, and then we will need to optimize the network parameters such as the weights and the biases to solve the problem of finding the optimal scheduling policy. In the average cost criterion the limit of the expected average cost over finite time intervals is minimized in a set of admissible policies. The control policy $f:\hat{E}\to A\left(x\right)$ is a stationary policy which prescribes the usage of a control action f(x)∈A(x) whenever at a decision epoch the system state is x∈E. The decision epochs arise whenever after serving any queue that queue becomes empty. For studied controllable queueing system operating under a control policy *f*, the average cost per unit of time for the ergodic system is of the form
(3)$$\begin{array}{c} g^{f}=\lim_{t\to \infty }\frac{1}{t}E^{f}\left[\int_{0}^{t}\sum_{i=1}^{N}c_{i}Q_{i}\left(u\right)du+\sum_{i=1}^{N}\sum_{j=1}^{N}c_{i,j}S_{i,j}\left(t\right)|X\left(0\right)=\left(d,0,\ldots ,0\right)\right], \end{array}$$
where $S_{i,j}\left(t\right)$ is the random number of switches from queue *i* to queue *j* in time interval [0,t]. Expectation $E^{f}$ must be calculated with respect to the control policy *f*. The policy $f^{*}$ is said to be optimal when for any admissible policy *f*,
(4)$$\begin{array}{c} g^{*}:=g^{f^{*}}=\min_{f}g^{f}. \end{array}$$

Our purpose focuses on a combination of simulation and neural network techniques. To verify the quality of results obtained by solving the optimization problem (Equation 4) we formulate an appropriate Markov decision problem. Then we compute the optimal control policy together with the corresponding average cost $g^{*}$ using a policy iteration algorithm, see, e.g., in Howard [30], Puterman [27], Tijms [28], which will be discussed in detail in a subsequent section.

## 3. Markov Decision Problem Formulation

Assume that the inter-arrival and service times are exponentially distributed, i.e., $\nu_{i}∼E\left(\lambda_{i}\right)$ and $\zeta_{i}∼E\left(\mu_{i}\right)$, i∈I. Under Markovian assumption the process (Equation 1) is a continuous-time Markov chain with a state space *E*. The MDP associated with this Markov process is represented as a five-tuple:(5)$$\begin{array}{c} (E,A,\left\{A\left(x\right),x\in E\right\},\lambda_{xy}\left(a\right),c\left(x,a\right)), \end{array}$$
where state space *E*, action spaces *A* and A(x) have been already defined in the previous section.

–$\lambda_{xy}$ is a transition rate to go from state *x* to state *y* by choosing a control action *a* is defined as
(6)$$\begin{array}{c} \lambda_{xy}\left(a\right)=\left\{\begin{array}{cc} \lambda_{i} & y=x+e_{i},\\ \mu_{i} & y=x-e_{i},\;d\left(x\right)=i,\;q_{i}\left(x\right)>1,\\ \mu_{i} & y=x-e_{i}+\left(a-i\right)e_{0},\;d\left(x\right)=i,\;q_{i}\left(x\right)=1,\;a\in A\left(x-e_{i}\right),\\ 0 & \mathrm{otherwise}\;\mathrm{for}\;y\neq x, \end{array}\right. \end{array}$$
where $\lambda_{xx}:=\lambda_{xx}\left(a\right)=-\sum_{y\neq x}\lambda_{xy}\left(a\right)$.–c(x,a) is an immediate cost in state x∈E by selecting an action *a*,
$$\begin{array}{c} c\left(x,a\right)=\sum_{i=1}^{N}c_{i}q_{i}\left(x\right)+\mu_{j}c_{j,a}1_{\left\{d\left(x\right)=j,q_{j}\left(x\right)=1\right\}}. \end{array}$$Here the first summand denotes the total holding cost of customers in all parallel queues in state *x* which is independent of a control action. Let $c\left(x\right)=\sum_{i=1}^{N}c_{i}q_{i}\left(x\right)$ and if $c_{i}=1$, i∈I, we get the number of customers in state *x*. The second summand includes the fixed cost $c_{j,a}$ for switching the server from the current queue *j* to the next queue with an index *a*.

The optimal control policy $f^{*}$ and the corresponding average cost $g^{f^{*}}$ are the solutions of the system of Bellman optimality equations,
(7)$$\begin{array}{c} Bv\left(x\right)=-\lambda_{xx}v\left(x\right)+g=\left[\sum_{i=1}^{N}\lambda_{i}+\mu_{j}1_{\left\{d\left(x\right)=j,q_{j}\left(x\right)\geq 1\right\}}\right]v\left(x\right)+g,\;x\in E, \end{array}$$
where *B* is a dynamic programming operator acting on value function v:E→R.

**Proposition** **1.**
*The dynamic programming operator B is defined as*

(8)
$$\begin{array}{cc} Bv(x) & =c\left(x\right)+\sum_{i=1}^{N}\lambda_{i}v\left(x+e_{i}\right)+\mu_{j}v\left(x-e_{j}\right)1_{\left\{d\left(x\right)=j,q_{j}\left(x\right)>1\right\}}\\ \; & +\mu_{j}\min_{a\in A(x-e_{j})}\left\{v\left(x-e_{j}+\left(a-j\right)e_{0}\right)+c_{j,a}\right\}1_{\left\{d\left(x\right)=j,q_{j}\left(x\right)=1\right\}},\;x\in E. \end{array}$$



**Proof.** From the Markov decision theory, e.g., [27,28], it is known that for continuous time Markov chain the operator *B* can be defined as $Bv\left(x\right)=\min_{a}\left[c\left(x,a\right)+\sum_{y\neq x}\lambda_{xy}v\left(y\right)\right]$. This equality for the proposed system can be obviously rewritten in form (Equation 8). In this equation, the first term c(x) represents the immediate holding cost of customers in state *x*. The second term by $\lambda_{i}$ describes the changes in value function due to new arrivals to the system. The third term by $\mu_{j}$ for $q_{j}\left(x\right)>1$ stands for the value function by service completion in the queue *j* where there are customers waiting for service. The last term by $\mu_{j}$ for $q_{j}\left(x\right)=1$ describes also a service completion which leads now to the state with an empty queue when a control action must be performed. Hence only the last term occurs with a min operator.    □

Note that the state space of the Markov decision model is countable infinite and the immediate costs c(x,a) are unbounded. The existence of the optimal stationary policy and convergence of the policy iteration algorithm can be verified for the system under study in a similar way as in Özkan and Kharoufeh [32], where first, the convergence of the value iteration algorithm for the equivalent discounted model is proved, and then, using the criteria proposed in Sennott [33], this result is extended to the policy iteration algorithm for the average cost criterion.

To solve Equation (Equation 8) in the policy iteration algorithm required to calculate the optimal control policy, we convert the multidimensional state space into a one-dimensional space by mapping $\Delta :E\to N_{0}$. The buffer sizes of the queues must be obviously truncated, namely $B_{i}<\infty$. Thereby the state $x=(d,q_{1},\ldots ,q_{N})$ can be rewritten in the following form:(9)$$\begin{array}{c} s:=\Delta \left(x\right)=d\left(x\right)\beta_{1,N}+\sum_{i=1}^{N}q_{i}\left(x\right)\beta_{i,N-1}, \end{array}$$
where $\beta_{i,j}=\prod_{k=i}^{j}\left(B_{k}+1\right)$ with $\beta_{N,N-1}=1$. The notation $\Delta^{-1}\left(s\right)$ will be used for the inverse function. In one-dimensional case the state transitions can be expressed as
$$\begin{array}{cc} \; & \Delta \left(x\pm e_{i}\right)=\Delta \left(x\right)\pm \beta_{i,N-1},\\ \; & \Delta \left(x+\left(a-j\right)e_{0}\right)=\Delta \left(x\right)+\left(a-j\right)\beta_{1,N}. \end{array}$$

The set of states *E* in truncated model is finite with a cardinality $\left|E\right|=N\beta_{1,N}$. The policy iteration Algorithm 1 consists of two main steps: Policy evaluation and policy improvement. In first step for the given initial control policy, it can be for example the LQF policy, the system of linear equations with constant coefficients must be solved. To make the system solvable the value function v(s) for one of the states can be assumed to be an arbitrary constant, e.g., v(0)=0 in the first state with d=1 and $q_{i}=0$. In this case we obtain from the optimality Equation (Equation 7) the equality $g=\sum_{i=1}^{N}\lambda_{i}v\left(\beta_{i,N-1}\right)$. The remaining equations can be solved numerically. As a solution we get the |E| values v(s) and the current value of the average cost *g*. In the policy improvement step, a control action *a* that minimizes the test value in the right-hand side of Equation (Equation 7) must be evaluated. The algorithm generates a sequence of control policies that converges to the optimum one. The convergence of the algorithm requires that the control actions in two adjacent iterations coincide in each state. To avoid policy improvement bouncing between equally good control actions in a given state, one can simply keep the previous control action unchanged if the corresponding test function is at least as large as for any other policy in determining the new policy. As an alternative to the proposed convergence criterion, one can use the values of average costs the variation of which should be for example less than a given some small value.

**Example** **1.**Consider the queueing system with N=4 queues. The buffer sizes are equal to $B_{i}=10$, i∈I. At these settings the number of states already reaches large values, |E|=58,564, which confirms one of significant restrictions on application of dynamic programming for this type of control problems. The switching costs can be defined for example as $c_{i,j}=j-i+4\;\mathrm{mod}\;4$. The holding costs $c_{i}$ for simplicity are assumed to be equal. The values of system parameters $\lambda_{i}$, $\mu_{i}$, $c_{i}$ and $c_{i,j}$ are summarized in Table 1 and reflect heterogeneity of the system parameters, i.e., $\lambda_{i}=0.05i$ and $\mu_{i}=\frac{3.750}{i}$.

**Algorithm 1** Policy iteration algorithm
1:**procedure** PIA($N,B_{i},\lambda_{i},\mu_{i},\;c_{i},\;c_{i,j},\;i,j\in I$)2:                                  ▹ Initial policy   $$\begin{array}{cc} \; & f^{\left(0\right)}\left(s\right)=\left\{\begin{array}{cc} \mathrm{Random}\{\underset{j\in I}{arg max}\left\{q_{j}\left(\Delta^{-1}\left(s\right)\right)\right\}\}\; & \;\mathrm{if}\;d\left(\Delta^{-1}\left(s\right)\right)=i\in I,\;q_{i}\left(\Delta^{-1}\left(s\right)\right)=0\\ 0 & \;\mathrm{otherwise} \end{array}\right. \end{array}$$3:  n←04:  $g^{\left(n\right)}\leftarrow \sum_{i=1}^{N}\lambda_{i}v^{\left(n\right)}\left(\beta_{i,N-1}\right)$                    ▹ Policy evaluation5:  **for** s=1to|E| **do**6:  

$$\begin{array}{cc} v^{\left(n\right)}\left(s\right) & \leftarrow \frac{1}{\sum_{i=1}^{N}\lambda_{i}+\mu_{j}1_{\left\{q_{j}\left(\Delta^{-1}\left(s\right)\right)>0\right\}}}[c\left(\Delta^{-1}\left(s\right)\right)+\mu_{j}c_{j,a}1_{\left\{d\left(\Delta^{-1}\left(s\right)\right)=j,q_{j}\left(\Delta^{-1}\left(s\right)\right)=1\right\}}-g^{\left(n\right)}\;\\ & +\sum_{i=1}^{N}\lambda_{i}\left[v^{\left(n\right)}\left(s+\beta_{i,N-1}\right)1_{\left\{q_{i}\left(\Delta^{-1}\left(s\right)\right)<B_{i}\right\}}+v\left(s\right)1_{\left\{q_{i}\left(\Delta^{-1}\left(s\right)\right)=B_{i}\right\}}\right]\;\\ & +\mu_{j}v^{\left(n\right)}\left(s-\beta_{j,N-1}\right)1_{\left\{d\left(\Delta^{-1}\left(s\right)\right)=j,q_{j}\left(\Delta^{-1}\left(s\right)\right)>1\right\}}\;\\ & +\mu_{j}v^{\left(n\right)}\left(s-\beta_{j,N-1}+\left(a-j\right)\beta_{1,N}\right)1_{\left\{d\left(\Delta^{-1}\left(s\right)\right)=j,q_{j}\left(\Delta^{-1}\left(s\right)\right)=1\right\}}],\\ a & \leftarrow f^{\left(n\right)}\left(s-\beta_{j,N-1}\right) \end{array}$$

7:  **end for**8:                              ▹ Policy improvement 

$$f^{\left(n+1\right)}\left(s\right)\leftarrow \underset{a\in A(s-\beta_{j,N-1})}{arg min}\left\{c_{j,a}+v^{\left(n\right)}\left(s-\beta_{j,N-1}+\left(a-j\right)\beta_{1,N}\right)\right\}1_{\left\{d\left(\Delta^{-1}\left(s\right)\right)=j,q_{j}\left(\Delta^{-1}\left(s\right)\right)=1\right\}}$$

9: **if** $f^{\left(n+1\right)}\left(s\right)\leftarrow f^{\left(n\right)}\left(s\right),\;s\in \left\{0,1,\ldots ,|E|\right\}$ **then return** $f^{\left(n+1\right)}\left(s\right),v^{\left(n\right)}\left(s\right),g^{\left(n\right)}$10:  **else**  n←n+1, **go to step 4**11:  **end if**12:
**end procedure**



These values correspond to the system load $\rho =\sum_{i=1}^{N}\rho_{i}=0.4$, that is the system is stable. This value is enough small to ensure on the one hand that the system is sufficiently loaded so that states appear where all queues are not empty, and on the other hand to minimize the probability of losing an arriving customer for given rather small buffer sizes. The solution of the large system of optimality equations is carried out numerically. The optimized average cost is $g^{*}=2.5632$.

Using Algorithm 1, we calculate the optimal scheduling policy. For some of states with fixed number of customers in the third and the fourth queues and varied number of customers in the first two queues the control actions are listed in Table 2. The first row of the table contains the values of the number of customers $q_{2}$ and $q_{1}$ in the second or first queue when a decision is made, respectively, when the first or second queue is emptied. The first column contains some selected states of the system for the fixed levels $q_{3}$ and $q_{4}$ of the third and fourth queues. As we can see, the optimal scheduling policy has a complex structure with a large number of thresholds, making it difficult to obtain any acceptable heuristic solution explicitly. To better visualise the complexity in structure of the optimal control policy, the background of the table cells changes in grey colour from darker to lighter backgrounds as the queue index decreases. The cμ-rule as expected is not optimal here, $g_{c\mu }=6.7237$ that is almost two and a half times more than the value of the average cost under the optimal policy. When the values $q_{1}$ and $q_{2}$ are small, the router schedules the server to serve the queues with low service rates. In this case the switching costs are low as well. According to the optimal scheduling policy the initiative to route a server to the queue with a higher service rate and switching costs increases as the length of the first two queues increases.

**Example** **2.**In this example we increase the arrival rates $\lambda_{i}$ as given in Table 3. The other parameters are fixed at the same values as in the previous example. The load factor now is ρ=0.64, and the corresponding optimized average cost is $g^{*}=3.8201$ and $g_{c\mu }=7.0420$.

The Table 4 of scheduling policy shows that as the system load increases the router switches the server to queue 2 or to queue 1 with a higher service rates at almost all queue lengths $q_{1}$ and $q_{2}$, respectively.

## 4. Event-Based Simulation for General Model

We use an event-based simulation to simulate the proposed queueing system. This technique is suitable for random process evaluation where it is sufficient to have the information about the time instants when changes in states occur. Such changes will be referred to as events. Note that although simulation modelling is extensively used in queueing theory, many papers lack explicitly described algorithms that readers can use for independent research. For more information on simulation methods with applications to single- and multi-server queueing systems, we can recommend Ebert et al. [34] and Franzl [35]. In this regard, it will certainly not be superfluous if we present and discuss here an algorithm for the system simulation which is not difficult to adapt for other similar systems.

In our case, the events are the arrivals to one of *N* parallel queues and the departures of customers from the queue *d* currently being served by the server. The present time is selected as a global time reference.

In Figure 2, on the time axis we mark the moments of arrival of new customers and the moments of their service in a fixed queue with index *d* by means of arrows above and below the axis, respectively. The dotted arrows indicate the arrival of new customers in other queues. The successive events are denoted by $\varepsilon_{i}$ and the corresponding time moments by $t(\varepsilon_{i})$. In the proposed queue simulation Algorithm 2 all the times are referred to the present time. Suppose that at the present moment of time there is a new arrival to the queue with the number *d*, which is serviced by the server, i.e., $t(\varepsilon_{i})=0$. Denote by $T_{x}\left(\varepsilon_{i}\right)$ the holding time of the system in state *x* up to the occurrence of the event $\varepsilon_{i}$. According to the time schema the holding time in a previous state is defined as $t_{i}=\min\left\{T_{x}\left(\varepsilon_{i}\right),T_{b}\left(d\right)-T_{x}\left(\varepsilon_{i-1}\right),\ldots \right\}=T_{x}\left(\varepsilon_{i}\right)$, where $T_{x}\left(\varepsilon_{i}\right)$ is a remaining inter-arrival time to the queue *d*, $T_{b}\left(d\right)$ stands for the generated service time after the event $\varepsilon_{i-2}$ of the previously occurred departure and the dots replace the time intervals associated with arrivals of customers in other queues. The next event is determined then by subtracting the holding time $t_{i}$ from the all event time intervals. In this case the current event is a new arrival. Thus, the holding time $t_{i+1}$ in state up to the event $\varepsilon_{i+1}$ of an arrival to some other queue which not equal to *d* is calculated by $t_{i+1}=\min\left\{T_{a}\left(d\right),T_{b}\left(d\right)-\sum_{j=i-1}^{i}T_{x}\left(\varepsilon_{j}\right),\ldots \right\}=T_{x}\left(\varepsilon_{i+1}\right)$. The subsequent holding times are calculated as follows, $t_{i+2}=\min\left\{T_{a}\left(d\right)-T_{x}\left(\varepsilon_{i+1}\right),T_{b}\left(d\right)-\sum_{j=i-1}^{i+1}T_{x}\left(\varepsilon_{j}\right),\ldots \right\}=T_{x}\left(\varepsilon_{i+2}\right)=T_{b}\left(d\right)-\sum_{j=i-1}^{i+1}T_{x}\left(\varepsilon_{j}\right)$, i.e., the event $\varepsilon_{i+2}$ is then the next departure from queue *d*, $t_{i+3}=\min\left\{T_{a}\left(d\right)-\sum_{j=i+1}^{i+2}T_{x}\left(\varepsilon_{j}\right),T_{b+1}\left(d\right),\ldots \right\}=T_{x}\left(\varepsilon_{i+3}\right)$, where $T_{b+1}\left(d\right)$ is the next generated service time, $t_{i+4}=\min\left\{T_{a}\left(d\right)-\sum_{j=i+1}^{i+3}T_{x}\left(\varepsilon_{j}\right),T_{b+1}\left(d\right)-T_{x}\left(\varepsilon_{i+3}\right),\ldots \right\}=T_{x}\left(\varepsilon_{i+4}\right)=T_{b+1}\left(d\right)-T_{x}\left(\varepsilon_{i+3}\right)$ and $t_{i+5}=\min\left\{T_{a}\left(d\right)-\sum_{j=i+1}^{i+4}T_{x}\left(\varepsilon_{j}\right),T_{b+2},\ldots \right\}=T_{x}\left(\varepsilon_{i}+5\right)=T_{a}\left(d\right)-\sum_{j=i+1}^{i+4}T_{x}\left(\varepsilon_{j}\right)$ is a remaining inter-arrival time for the next arrival to the queue *d*. Continuing the process in a similar manner, all holding times of the system in the corresponding states are evaluated. By summing up the times $t_{i}$ we obtain the total simulation running time of the system simT. The average cost per unit of time is then obtained by division of the accumulated cost by the time symT.

The time instants of arrival events to the queue q∈I are stored in vector variable $T_{a}$ and the departure events in the queue with a number *q* in $T_{b}\left[q\right]$. The Algorithm 2 contains pseudo-code of the main elements of the event based simulation procedure.
**Algorithm 2** Queue simulation algorithm1:**procedure** QSIM($N,B_{i},A_{i},B_{i},\;c_{i},\;c_{i,j},\;i,j\in I$, θ, $n_{\max}$,$n_{\min}$)                          ▹ Initialization2:    $T_{a}=\left(0,\ldots ,0\right),\;|T_{a}|=N,\;T_{b}=\left(\left(\infty \right),\ldots ,\left(\infty \right)\right),\;\left|T_{b}\right|=N,\;xT=0,\;i=0,\;sc=0$3:    d=Random[{1,…,N}],x=(d,0,…,0),|x|=N+14:    **while** $i<n_{\max}$ **do**                                                                                      ▹ State recording5:        $t_{i}\leftarrow \min\left(T_{a},\min\left(T_{b}\left[1\right]\right),\ldots ,\min\left(T_{b}\left[N\right]\right)\right)$6:        $T_{a}\leftarrow T_{a}-t_{i}$7:        **for** q=1toN **do**8:             $T_{b}\left[q\right](2:|T_{b}\left[q\right]|)\leftarrow T_{b}\left[q\right](2:|T_{b}\left[q\right]\left|\right)-t_{i}$9:        **end for**10:        **if** $i>n_{\min}$ **then**11:           $simT\leftarrow simT+t_{i}$                                                                             ▹ Simulation time12:           $xT\leftarrow xT+t_{i}\sum_{j=1}^{N}c_{j}x\left[j+1\right]+sc$                                                  ▹ Sum up the cost13:        **end if**14:        cs←015:        **for** q=1toN **do**16:           **if** $\left(q=d\;\&\;T_{a}\left[q\right]\leq \varepsilon \right)$ **then return**17:               $T_{a}\left[q\right]\leftarrow \mathrm{RandomVariate}\left[A_{q}\left(t\right)\right]$                              ▹ Generate interarrival time18:               $x\leftarrow x+e_{q+1}\left(N+1\right)$, i←i+119:               **if** $|T_{b}\left[q\right]|\leq 1$ **then return**20:                   $T_{b}\left[q\right]\leftarrow \left(T_{b}\left[q\right],\mathrm{RandomVariate}\left[B_{q}\left(t\right)\right]\right)$                ▹ Generate service time21:               **end if**22:           **end if**23:           **if** $\left(q=d\;\&\;T_{a}\left[q\right]>\varepsilon \;\&\;|T_{b}\left[q\right]\leq \varepsilon |>0\right)$ **then return**24:               $\mathrm{index}\leftarrow T_{b}\left[q\right]\leq \varepsilon$                                                  ▹ Index of the current departure25:               $T_{b}\left[q\right]\leftarrow \left(T_{b}\left[q\right]\setminus T_{b}\left[q\right]\left[\mathrm{index}\right]\right)$                              ▹ Remove current departure26:               $x\leftarrow x-e_{q+1}\left(N+1\right)$27:               **if** x[q+1]≥1 **then return**28:                   $T_{b}\left[q\right]\leftarrow \left(T_{b}\left[q\right],\mathrm{RandomVariate}\left[B_{q}\left(t\right)\right]\right)$29:               **end if**30:               **if** x[q+1]=0 **then return**31:                   a←f(x,θ), d←a                                                            ▹ New server scheduling32:                   $x\leftarrow x+\left(a-q\right)e_{1}\left(N+1\right)$33:                   $sc=c_{q,a}$34:                   **if** x[a+1]>0 **then return**35:                       $T_{b}\left[a\right]\leftarrow \left(T_{b}\left[a\right],\mathrm{RandomVariate}\left[B_{a}\left(t\right)\right]\right)$             ▹ Generate service time36:                   **end if**37:               **end if**38:           **end if**39:           **if** $\left(q\neq d\;\&\;T_{a}\left[q\right]\leq \varepsilon \right)$ **then return**40:               $T_{a}\left[q\right]\leftarrow \mathrm{RandomVariate}\left[A_{q}\left(t\right)\right]$                                  ▹ Generate inter-arrival time41:               $x\leftarrow x+e_{q+1}\left(N+1\right)$, i←i+142:           **end if**43:        **end for**44:    **end while**45:    g←xT/simT46:**end procedure**

## 5. Neural Network Architecture

In our model, we propose to equip the router with a trained neural network. This network will determine an index of the queue that the server will serve next based on the information about the system state at a decision epoch when the server finishes service of the current queue. We have chosen a simple architecture for the neural network consisting of only two layers in such a way that, on the one hand, it would have a small number of parameters for further optimization and, on the other hand, that the quality of correct classification of some fixed initial control policy would be equal to at least 95%. The proposed neural network has one linear layer which represents an affine transformation and softmax normalization layer as illustrated in Figure 3.

The input includes N+1 neurons according to the system state $x=(d,q_{1},\ldots ,q_{N})$, where $q_{d}\left(x\right)=0$. The neuron 0 gets the information on d(x), the *i*th neuron for i∈I gets the information on the state of *i*th queue. When the server finishes service at queue *d*, then the neural network classifies this state to one of *N* classes which defines a current control action a∈A in state *x*. The hidden linear layer consists of *N* neurons $y={\left(y_{1},\ldots ,y_{N}\right)}^{\prime }$ which are connected with an input neurons via the system of linear equations
$$\begin{array}{cc} \; & y_{1}=w_{1,0}x_{0}+w_{1,1}x_{1}+\cdots +w_{1,N}x_{N}+b_{1}\\ \; & y_{2}=w_{2,0}x_{0}+w_{2,1}x_{1}+\cdots +w_{2,N}x_{N}+b_{2}\\ \; & \ldots \\ \; & y_{N}=w_{N,0}x_{0}+w_{N,1}x_{1}+\cdots +w_{N,N}x_{N}+b_{N}, \end{array}$$
or in matrix form y=Wx+B with $W\in R^{N\times (N+1)}$ and $B\in R^{N}$, where
(10)$$\begin{array}{c} W=\left(\begin{array}{cccc} w_{1,0} & w_{1,1} & \ldots & w_{1,N}\\ w_{2,0} & w_{2,1} & \ldots & w_{2,N}\\ \vdots & \vdots & \ddots & \vdots \\ w_{N,0} & w_{N,1} & \ldots & w_{N,N} \end{array}\right)=\left(\begin{array}{c} w_{1}\\ w_{2}\\ \vdots \\ w_{N} \end{array}\right)\;\;\mathrm{and}\;\;B={\left(b_{1},b_{2},\ldots ,b_{N}\right)}^{\prime } \end{array}$$
with $w_{i}=\left(w_{i,0},w_{i,1},\ldots ,w_{i,N}\right)$ are, respectively, the matrix of weights and the vector of biases of the given neural network which must be estimated by means of the training set. The softmax layer z=softmax(y) is a final layer of the multiclass classification. The softmax layer generates as an output the vector of *N* estimated probabilities of the input sample $y_{i}$, where the *i*th entry is the likelihood that *x* belongs to class *i*. The vector *y* is normalized by the transformation
$$\begin{array}{c} z=\left(\begin{array}{c} z_{1}\\ z_{2}\\ \vdots \\ z_{N} \end{array}\right)=\frac{1}{\sum_{i=1}^{N}e^{y_{i}}}\left(\begin{array}{c} e^{y_{1}}\\ e^{y_{2}}\\ \vdots \\ e^{y_{N}} \end{array}\right). \end{array}$$
The class number is then defined as $\hat{a}=arg maxz_{i}$. Hence, the output *z* is a mapping of the form z=φ(x,θ), where $\theta \in R^{N(N+2)}$ is the parameter vector of the neural network which includes all entries of the weight matrix $W\in R^{N\times (N+1)}$ and the bias vector $B\in R^{N}$, i.e.,
(11)$$\begin{array}{c} \theta =(w_{1},w_{2},\ldots ,w_{N},B^{\prime }). \end{array}$$
The values of the parameter vector θ of the initial control policy, which in the next section will be used as a starting solution for optimization procedure, are obtained by training the neural network on some known heuristic control policy. In our case this policy is the LQF. In the training phase the following optimization problem must be solved given the training set ${\left\{x^{\left(k\right)}\right\}}_{k=1}^{m}\to {\left\{a^{\left(k\right)}\right\}}_{k=1}^{m}$,
(12)$$\begin{array}{c} \theta^{*}=\underset{\theta }{arg min}\frac{1}{m}\sum_{k=1}^{m}l_{k}\left(\theta \right), \end{array}$$
where a non-negative loss function
$$l_{k}\left(\theta \right)=-\sum_{i=1}^{N}1_{\left\{a^{\left(k\right)}=i\right\}}\ln z_{i}^{\left(k\right)}$$
with $z_{i}^{\left(k\right)}=P\left[a^{\left(k\right)}=i|x^{\left(k\right)},\theta \right]$ takes the value 0 only if the class of the *k*th element of a sample is *i*, i.e., $\hat{a}=a^{\left(k\right)}$. The problem (Equation 12) can be solved in a usual way by the stochastic gradient descent method, where a single learning rate η to update all parameters is maintained. The corresponding iterative expression is given below,
$$\theta^{\left(n\right)}=\theta^{\left(n-1\right)}-\eta \nabla_{\theta }\left(\frac{1}{m}\sum_{k=1}^{m}l_{k}\left(\theta^{\left(n-1\right)}\right)\right),$$
where $\nabla_{\theta }$ is a Nabla-operator defining the gradient of the function relative to the parameter vector θ. In our calculations we use the adaptive moment estimation algorithm (ADAM) to solve the problem (Equation 12). It updates iteratively the parameters of the neural network based on training data. The ADAM calculates independent adaptive learning rates for the elements of θ by evaluating the first-moment and second moment estimation of the gradient. The method is simple to implement, computationally efficient, requires little memory and is invariant to diagonal changes in gradients. The further detailed information regarding ADAM algorithm can be found in Kingma and Ba [36]. Despite the fact that the ADAM algorithm can be found across various sources, we have also chosen to cite it in this article. The main steps required for the iterative updating the parameter vector θ are summarized in the Algorithm 3.

The parameters of the Algorithm 3 are fixed to η=0.001, $\beta_{1}=0.9$, $\beta_{2}=0.999$, $\varepsilon =10^{-8}$ and δ=0.001. The classification accuracy of the proposed neural network trained on the LQF policy is over 97%. The test phases of the trained network were conducted on system states with a queue length of up to 100 customers per queue. Thus, this starting network can be used to generate control actions of the initial control policy for subsequent parameters’ optimization of this neural network.

**Algorithm 3** Adaptive moment estimation algorithm
1:**procedure** ADAM(η,$\beta_{1}$,$\beta_{2}$,ε,δ)2:  $M_{1}^{\left(0\right)}\leftarrow \left(0,\ldots ,0\right)$                 ▹ Initialisation of the moment 13:  $M_{2}^{\left(0\right)}\leftarrow \left(0,\ldots ,0\right)$                 ▹ Initialisation of the moment 24:  CI←0                        ▹ Convergence index5:  n←06:  **while** CI=0 **do**7:   n←n+18:   $G^{\left(n\right)}\leftarrow \nabla_{\theta }\left(\frac{1}{m}\sum_{k=1}^{m}l_{k}\left(\theta^{\left(n-1\right)}\right)\right)$         ▹ Calculate the gradient at step *n*9:   $M_{1}^{\left(n\right)}\leftarrow \beta_{1}M_{1}^{\left(n-1\right)}+\left(1-\beta_{1}\right)G^{\left(n\right)}$       ▹ Update the biased first moment10:   $M_{2}^{\left(n\right)}\leftarrow \beta_{2}M_{2}^{\left(n-1\right)}+\left(1-\beta_{2}\right){\left(G^{\left(n\right)}\right)}^{2}$     ▹ Update the biased second moment11:   ${\hat{M}}_{1}^{\left(n\right)}\leftarrow \frac{M_{1}^{\left(n\right)}}{1-\beta_{1}^{n}}$               ▹ The bias-corrected first moment12:   ${\hat{M}}_{2}^{\left(n\right)}\leftarrow \frac{M_{2}^{\left(n\right)}}{1-\beta_{2}^{n}}$             ▹ The bias-corrected second moment13:   $\theta^{\left(n\right)}=\theta^{\left(n-1\right)}-\eta \frac{{\hat{M}}_{1}^{\left(n\right)}}{\sqrt{{\hat{M}}_{2}^{\left(n\right)}+\varepsilon }}$           ▹ Update the parameter vector14:   **if** $|\theta^{\left(n\right)}-\theta^{\left(n-1\right)}|<\delta$ **then return** $\theta^{\left(n\right)}$15:      CI←1                     ▹ Check the convergence16:   **end if**17:  **end while**18:
**end procedure**



## 6. Optimization of the Neural-Network-Based Scheduling Policy

Denote by θ the known parameter vector of the trained neural network as was defined in (Equation 11). The function g(θ) means the average cost for the queueing system where the router chooses an action obtained from the trained neural network with the parameter vector θ. We adapt further a simulated annealing method described in Algorithm 4 for discrete stochastic optimization of the average cost function
(13)$$\begin{array}{c} g^{*}=\min_{\theta }g\left(\theta \right),\;\theta^{*}=\underset{\theta }{arg min}g\left(\theta \right) \end{array}$$
with a multidimensional parameter vector θ. This algorithm is quite straightforward. It needs some starting solution and in each iteration the algorithm evaluates for the randomly selected neighbor values of the function parameters the corresponding function value. If the neighbor occurs to be better than the current solution with respect to value of the objective function, algorithms replaces the current solution with a new one. If the neighbor value is worse, the algorithm keeps the current solution with a high probability and chooses a new value with a specified low probability.

The simulated annealing requires the finite discrete space for the parameters of the optimized function. It is assumed that all weights and biases of the neural network summarized in the vector θ take values in the interval $\left[\theta_{\min},\theta_{\max}\right]$ with a low bound $\theta_{\min}$ and an upper bound $\theta_{\max}$. Moreover, this interval is quantized in such a way that $\theta_{i}$, i=1,…,N(N+2), takes only discrete values $\theta_{\min}+k\Delta$, k=0,1,…,Q, where $Q=\frac{\theta_{\max}-\theta_{\min}}{\Delta }$ is a quantization level. Note that the domains for the elements of the parameter vector θ can be specified separately, and the values of the vector obtained by training the neural network based on the optimal policy of the Markov model will be suitable for determining the possible maximum and minimum bounds. In this case it is possible to achieve faster convergence of Algorithm 4 to the optimal value.

**Algorithm 4** Simulated annealing algorithm
1:**procedure** SA(T(n),Δ,*m*,η,τ,ν,$\theta_{\min}$,$\theta_{\max}$)                                                           ▹ Initialisation2:    
$\theta^{\left(0\right)}\leftarrow \left(w_{1,\mathrm{LQF}},w_{2,\mathrm{LQF}},\ldots ,w_{N,\mathrm{LQF}},B_{\mathrm{LQF}}^{\prime }\right)$3:    n←04:    $\bar{g}\left(\theta^{\left(n\right)}\right)\leftarrow \frac{1}{m}\sum_{k=1}^{m}\mathrm{QSIM}\left(\ldots ,\theta^{\left(n\right)}\right)$5:    $g^{*}\leftarrow \bar{g}\left(\theta^{\left(n\right)}\right)$, $\theta^{*}\leftarrow \theta^{\left(n\right)}$6:    **while** T(n)>τ||n<ν **do**7:          n←n+1                                                                                                        ▹ Perturbation8:          i←Random[{1,…,N(N+2)}]9:          $\xi \leftarrow \mathrm{Random}[\left\{\max\left\{-\eta \Delta ,\theta_{\min}-\theta_{i}^{\left(n-1\right)}\right\},\ldots ,\min\left\{\eta \Delta ,\theta_{\max}-\theta_{i}^{\left(n-1\right)}\right\}\right\}]$10:        $\theta^{\left(n\right)}\leftarrow \theta^{\left(n-1\right)}+\xi e_{i}^{\prime }$11:        $\bar{g}\left(\theta^{n}\right)\leftarrow \frac{1}{m}\sum_{k=1}^{m}\mathrm{QSIM}\left(\ldots ,\theta^{\left(n\right)}\right)$                                                                ▹ Acceptance12:        **if** $\bar{g}\left(\theta^{\left(n\right)}\right)-g^{*}-S_{g\left(\theta^{\left(n\right)}\right),g\left(\theta^{\left(n-1\right)}\right)}t_{2m-2;1-\alpha }>0$ **then return**13:           $p\leftarrow e^{-\frac{\bar{g}\left(\theta^{\left(n\right)}\right)-g^{*}-S_{g\left(\theta^{\left(n\right)}\right),g\left(\theta^{\left(n-1\right)}\right)}t_{2m-2;1-\alpha }}{T(n)}}$14:        **else**   p←115:        **end if**16:        u←Random[]17:        **if** p≥u** then return **$g^{*}\leftarrow \bar{g}\left(\theta^{\left(n\right)}\right)$, $\theta^{*}\leftarrow \theta^{\left(n\right)}$18:        **else**   $\theta^{\left(n\right)}\leftarrow \theta^{\left(n-1\right)}$, m←m+119:        **end if**20:    **end while**21:
**end procedure**



Since the average cost function *g* can not be calculated analytically, for this purpose a simulation technique is used. As shown in Algorithm 4, at each iteration at the step where the current solution can be accepted with a given probability we need to calculate the difference between the object functions. Due to the fact that this function can only be calculated numerically, it is necessary to check whether this difference is statistically significant at each iteration of the algorithm. The algorithm is modified in such a way that the *t*-test for two samples is used to compare the expected values of two normally distributed samples with unknown but equal variances. Denote by $\theta_{1}$ and $\theta_{2}$, respectively, the current and the modified parameter vector and by
(14)$$\begin{array}{c} \bar{g}\left(\theta_{1}\right)=\frac{1}{m}\sum_{k=1}^{m}g^{\left(k\right)}\left(\theta_{1}\right),\;\bar{g}\left(\theta_{2}\right)=\frac{1}{m}\sum_{k=1}^{m}g^{\left(k\right)}\left(\theta_{2}\right) \end{array}$$
two corresponding first empirical moments of the objective function. According to the *t*-test the null hypothesis which states that for the modified vector the average cost is statistically smaller then the previous solution is rejected if
(15)$$\begin{array}{c} \bar{g}\left(\theta_{2}\right)-\bar{g}\left(\theta_{1}\right)-S_{g\left(\theta_{1}\right),g\left(\theta_{2}\right)}t_{2m-2;1-\alpha }>0, \end{array}$$
where $t_{m;q}$ stands for the *q*-quantile of the *t*-distribution and statistics $S_{g\left(\theta_{1}\right),g\left(\theta_{2}\right)}$ is defined as
(16)$$\begin{array}{c} S_{g\left(\theta_{1}\right),g\left(\theta_{2}\right)}=\sqrt{\frac{V_{g(\theta_{1})}^{\left(m\right)}+V_{g(\theta_{2})}^{\left(m\right)}}{m}}, \end{array}$$
with empirical variances $V_{g(\theta_{1})}^{\left(m\right)}$ and $V_{g(\theta_{2})}^{\left(m\right)}$.

Below, we briefly describe the main steps of the Algorithm 4. At the initialisation step of the algorithm, the neural network is trained based on the LQF control policy. The parameter vector is then equal to the initial vector $\theta^{\left(0\right)}$ to be optimized. The simulation Algorithm 2 is then used to calculate the initial sample ${\left\{g^{\left(k\right)}\left(\theta^{\left(0\right)}\right)\right\}}_{k=1}^{m}$ with $g^{\left(k\right)}\left(\theta^{\left(0\right)}\right)=\mathrm{QSIM}\left(\ldots \right)$ of the average cost function for a given initial parameter vector $\theta^{\left(0\right)}$ and the corresponding first empirical moment $\bar{g}\left(\theta^{\left(0\right)}\right)$. These values are set as the current solution $g^{*}$ and $\theta^{*}$ to the optimization problem (Equation 13). At the perturbation step, a randomly chosen element of the previous parameter vector $\theta^{\left(n-1\right)}$ must be randomly perturbed on the specified set
$$L\left(i\right)=\{\max\left\{\theta_{i}^{\left(n-1\right)}-\eta \Delta ,\theta_{\min}\right\},\ldots ,\min\left\{\theta_{i}^{\left(n-1\right)}+\eta \Delta ,\theta_{\max}\right\}\}$$
of admissible discrete domain. For a new parameter vector $\theta^{\left(n\right)}$ next sample ${\left\{g^{\left(k\right)}\left(\theta^{\left(n\right)}\right)\right\}}_{k=1}^{m}$ of average costs must be calculated together with the first empirical moment $\bar{g}\left(\theta^{\left(n\right)}\right)$. At the acceptance step, a new policy $\theta^{\left(n\right)}$ can be accepted as a current solution with a probability *p* defined as
$$\begin{array}{c} p=\left\{\begin{array}{cc} 1 & \;\mathrm{if}\;\bar{g}\left(\theta^{\left(n\right)}\right)\leq g^{*}\\ e^{-\frac{\bar{g}\left(\theta^{\left(n\right)}\right)-g^{*}-S_{g\left(\theta^{\left(n\right)}\right),g\left(\theta^{\left(n-1\right)}\right)}t_{2m-2;1-\alpha }}{T(n)}} & \;\mathrm{if}\;\bar{g}\left(\theta^{\left(n\right)}\right)>g^{*}, \end{array}\right. \end{array}$$
where T(n) is the temperature at the *n*th iteration. If a new policy $\theta^{\left(n\right)}$ is accepted, then it is defined together with a corresponding average cost $\bar{g}\left(\theta^{\left(n\right)}\right)$ as a current solution. Otherwise, the last change in the parameter vector $\theta^{\left(n-1\right)}$ must be reversed, i.e., $\theta^{\left(n\right)}=\theta^{\left(n-1\right)}$ and the sample size *m* for calculating the first moments is updated. Then the perturbation step must be repeated. For termination of the algorithm the stopping criteria T(n)<τ or n<ν is used.

We note that the classical simulated annealing method generates for some function g(θ) a sample $\theta^{\left(n\right)}$ which for the constant temperature T(n)=T can be interpreted as a realization of a homogeneous Markov chain $\left\{\Theta_{n}\right\}\{n\in N_{0}\}$ with transition probabilities
(17)$$\begin{array}{c} p_{\theta_{i},\theta_{j}}=P\left[\Theta_{n+1}=\theta_{j}|\Theta_{n}=\theta_{i}\right]=\frac{1}{\left|L(i)\right|}P\left[U_{n}\leq e^{-\frac{g\left(\theta_{j}\right)-g\left(\theta_{i}\right)}{T}}\right],\;\theta_{j}\in L\left(i\right), \end{array}$$
where $U_{n}$ is a uniformly distributed random variable on the interval [0,1]. It is easy to show that the modified transition probabilities, where the objective function is calculated numerically, converges to the transition probabilities (Equation 17) which in turn can guarantee the convergence to an optimal solution.

**Proposition** **2.**
*The acceptance probability p(n) satisfies the limit relation*

(18)
$$\begin{array}{c} \lim_{n\to \infty }p\left(n\right)=\lim_{n\to \infty }P\left[U_{n}\leq e^{-\frac{\bar{g}\left(\theta_{j}\right)-\bar{g}\left(\theta_{i}\right)-S_{g\left(\theta_{j}\right),g\left(\theta_{i}\right)}t_{2m-2;1-\alpha }}{T}}\right]=P\left[U_{n}\leq e^{-\frac{g\left(\theta_{j}\right)-g\left(\theta_{i}\right)}{T}}\right]. \end{array}$$



**Proof.** The probability $P[U_{n}\leq X]$ can be obviously rewritten as
$$\begin{array}{c} P\left[U_{n}\leq X\right]=\int_{0}^{1}P\left[u\leq X\right]f_{U_{n}}\left(u\right)du=E\left[X\right], \end{array}$$
where $X=e^{-\frac{\bar{g}\left(\theta_{j}\right)-\bar{g}\left(\theta_{i}\right)-S_{g\left(\theta_{j}\right),g\left(\theta_{i}\right)}t_{2m-2;1-\alpha }}{T}}$. Then the following relation holds,
$$\begin{array}{c} \lim_{n\to \infty }E\left[e^{-\frac{\bar{g}\left(\theta_{j}\right)-\bar{g}\left(\theta_{i}\right)-S_{g\left(\theta_{j}\right),g\left(\theta_{i}\right)}t_{2m-2;1-\alpha }}{T}}\right]=E\left[e^{-\frac{g\left(\theta_{j}\right)-g\left(\theta_{i}\right)}{T}}\right], \end{array}$$
due to the strong law of large numbers and the fact that for n→∞ the sample size m→∞ and hence
$$\lim_{m\to \infty }S_{g\left(\theta_{j}\right),g\left(\theta_{i}\right)}=\lim_{m\to \infty }\sqrt{\frac{\sigma_{j}^{2}+\sigma_{i}^{2}}{m}}=0.$$□

## 7. Numerical Analysis

Consider the queueing system with N=4. We first analyse a Markov model, where the parallel queues are of the type M/M/1 with $\nu_{i}∼E\left(\lambda_{i}\right)$ and $\zeta_{i}∼E\left(\mu_{i}\right)$, i∈I, the coefficient of variation $CV_{\nu_{i}}^{2}=CV_{\zeta_{i}}^{2}=1$. The values of system parameters $\lambda_{i}$ and $\mu_{i}$ are fixed as in examples 1 and 2 which will refer to as Cases 1 and 2. We compare the optimization results obtained by combining the simulation, neural network and simulated annealing algorithm with the results evaluated by the policy iteration algorithm. In Cases 1 and 2, the weights and the biases of the neural network trained on the calculated by PIA optimal scheduling policy take, respectively, the following values
$$\begin{array}{cc} \; & W_{\mathrm{PIA}}=\left(\begin{array}{ccccc} 0.4 & 3.2 & 0.3 & 0.1 & 0.2\\ -0.3 & -3.8 & 0.8 & 0.2 & 0.2\\ 0.1 & -2.9 & -3.6 & 0.4 & 0.3\\ 0.4 & -0.3 & -1.6 & -1.3 & 0.3 \end{array}\right)\;B_{\mathrm{PIA}}=\left(-1.6,1.0,0.9,0.4\right),\\ \; & W_{\mathrm{PIA}}=\left(\begin{array}{ccccc} 0.5 & 2.0 & 0.2 & 0.0 & 0.3\\ -0.3 & -2.0 & 0.7 & 0.0 & 0.3\\ 0.2 & -1.3 & -2.1 & 0.0 & 0.4\\ 0.1 & 0.0 & -1.0 & 0.0 & 0.3 \end{array}\right)\;\;\;\;\;B_{\mathrm{PIA}}=\left(-1.1,1.1,0.6,0.0\right). \end{array}$$

On the basis of these values, we can estimate in the simulation annealing Algorithm 4 the domain or solution space for each element of the vector θ. For simplicity, in our experiments we set common boundaries for all elements as $\theta_{\min}=-6$ and $\theta_{\max}=6$. The length of the increment Δ=0.1 implies the quantization level Q=120. Next, we set η=6, ν=200, and $T\left(n\right)=\frac{0.2}{\log(n)}$. As an initial vector $\theta^{\left(0\right)}$ we take the parameter vector obtained by training the neural network on the LQF policy. For the initial control policy, one could also choose the policy $W_{\mathrm{PIA}},B_{\mathrm{PIA}}$ obtained by Algorithm 1. However, we would like to check the convergence of the algorithm when choosing not the best initial solution, since in general case one usually chooses either some heuristic policy or an arbitrary one. The empiric average cost $\bar{g}\left(\theta^{\left(n\right)}\right)$ for each iteration step is calculated based on sample with a size m≥20. The accumulation of sample data in QSIM Algorithm 2 is carried out after 1000 customers have entered the system and is completed after 5000 customers have entered the system.

Application of the Algorithm 4 to a Markov model leads to the following optimal solutions:Case 1:Optimal solution is reached at n=184, $g^{*}=g\left(\theta^{*}\right)=2.2436$,
$$\begin{array}{cc} \; & W_{\mathrm{LQF}}=\left(\begin{array}{ccccc} 0.5 & 2.0 & -1.0 & 0.7 & -0.9\\ 0.3 & -0.7 & 1.6 & -0.7 & -0.9\\ 0.4 & -0.6 & -1.3 & 2.0 & -0.8\\ 0.0 & -0.6 & -1.3 & -1.8 & 1.9 \end{array}\right)\Rightarrow W_{\mathrm{SA}}=\left(\begin{array}{ccccc} 0.7 & 5.3 & -0.3 & -0.8 & 0.9\\ 0.4 & -2.8 & 1.6 & 0.2 & -0.1\\ 0.4 & -5.7 & -5.9 & 1.6 & 0.9\\ 1.0 & -0.7 & -2.3 & -2.8 & 1.2 \end{array}\right)\\ \; & \;B_{\mathrm{LQF}}={\left(0.5,-0.1,-0.2,-0.2\right)}^{\prime }\;\;\Rightarrow B_{\mathrm{SA}}={\left(-1.8,-0.2,-0.6,-0.3\right)}^{\prime }. \end{array}$$Case 2:Optimal solution is reached at n=188, $g^{*}=g\left(\theta^{*}\right)=3.2279$,
$$\begin{array}{cc} \; & W_{\mathrm{LQF}}=\left(\begin{array}{ccccc} 0.5 & 2.0 & -1.0 & 0.7 & -0.9\\ 0.3 & -0.7 & 1.6 & -0.7 & -0.9\\ 0.4 & -0.6 & -1.3 & 2.0 & -0.8\\ 0.0 & -0.6 & -1.3 & -1.8 & 1.9 \end{array}\right)\Rightarrow W_{\mathrm{SA}}=\left(\begin{array}{ccccc} 0.4 & 5.3 & -0.6 & 0.8 & 0.4\\ -0.2 & -3.2 & 1.8 & 0.0 & 0.3\\ 0.5 & -3.1 & -3.9 & 1.4 & 0.0\\ 0.4 & -1.0 & -1.0 & -3.5 & 1.3 \end{array}\right)\\ \; & \;B_{\mathrm{LQF}}={\left(0.5,-0.1,-0.2,-0.2\right)}^{\prime }\;\;\Rightarrow B_{\mathrm{SA}}={\left(-2.5,0.6,0.0,0.6\right)}^{\prime }. \end{array}$$

We see that the elements of matrices $W_{\mathrm{PIA}}$ and $W_{\mathrm{SA}}$ are different, but they are markedly similar in terms of the elements with dominant values. The optimization process of the scheduling policy is illustrated in Figure 4. In addition to values of the average cost function obtained at each iteration step of the simulated annealing algorithm, the figures show horizontal dotted and dash-dotted lines, respectively, at level of the average cost $g_{\mathrm{LQF}}=9.7093$ and $g_{c\mu }=4.1984$ in figure labelled by (a) and $g_{\mathrm{LQF}}=11.1740$ and $g_{c\mu }=5.2546$ in figure labelled by (b) for the LQF and cμ heuristic policies. As expected, a non-optimal control policy LQF implies too high average cost. The results look much better for policy cμ, but still the presence of switching costs significantly worsens the performance of this policy. The red horizontal line indicates the average cost $g_{\mathrm{PIA}}=2.5632$ and $g_{\mathrm{PIA}}=3.5500$ obtained by solving the Markov decision problem using the policy iteration Algorithm 1. We can observe that the values are quite close to those obtained by random search. However, some small difference may be due, firstly, to the fact that the simulation is used for calculations and the results have a certain scattering, and, secondly, we do not exclude the influence of boundary states in the Markov model, where a buffer size truncation has been used. Testing the hypothesis for the difference between the optimal average costs $g^{*}$ and $g_{\mathrm{PIA}}$ at least for our model showed the values to be statistically equivalent. In the figures, we have also marked with triangles those iteration steps with accepted policy (AP) where the perturbed parameter vector has been accepted. The number of accepted points in Case 1 and 2 is equal, respectively, to 98 and 110. From above results in case of exponential time distributions we can make the following observations. If the parameter vector $\theta^{\left(0\right)}$ with elements $W_{\mathrm{PIA}}$ and $B_{\mathrm{PIA}}$ is used for the initial scheduling policy, then one can expect the faster convergence of the simulated annealing algorithm to the optimal solution which was confirmed numerically. If an optimal policy for a controlled Markov process is not available, e.g., when the number of queues is too large, in this case it is reasonable to use the static cμ-rule as an initial policy.

Figure 5 displays experiments realized for the queues of the type D/D/1 with deterministic inter-arrival and service times which are equal to corresponding mean values $\frac{1}{\lambda_{i}}$ and $\frac{1}{\mu_{i}}$ of the Markov model. Here the coefficient $CV_{\nu_{i}}^{2}=CV_{\zeta_{i}}^{2}=0$. The SA algorithm converges to the values $g^{*}=1.6500$ and $g^{*}=2.0326$, respectively, for Case 1 and 2 with the following optimal policies,

Case 1:

$$\begin{array}{cc} \; & W_{\mathrm{LQF}}=\left(\begin{array}{ccccc} 0.5 & 2.0 & -1.0 & 0.7 & -0.9\\ 0.3 & -0.7 & 1.6 & -0.7 & -0.9\\ 0.4 & -0.6 & -1.3 & 2.0 & -0.8\\ 0.0 & -0.6 & -1.3 & -1.8 & 1.9 \end{array}\right)\Rightarrow W_{\mathrm{SA}}=\left(\begin{array}{ccccc} 0.4 & 2.5 & -0.7 & -0.3 & -0.2\\ 0.4 & -3.6 & 1.6 & 0.5 & -0.5\\ 0.5 & -5.8 & -1.2 & 1.2 & 0.0\\ 0.9 & -0.1 & -3.5 & -2.9 & 1.2 \end{array}\right)\\ \; & \;B_{\mathrm{LQF}}={\left(0.5,-0.1,-0.2,-0.2\right)}^{\prime }\;\;\;\Rightarrow B_{\mathrm{SA}}={\left(0.1,0.5,0.8,0.6\right)}^{\prime }. \end{array}$$

Case 2:

$$\begin{array}{cc} \; & W_{\mathrm{LQF}}=\left(\begin{array}{ccccc} 0.5 & 2.0 & -1.0 & 0.7 & -0.9\\ 0.3 & -0.7 & 1.6 & -0.7 & -0.9\\ 0.4 & -0.6 & -1.3 & 2.0 & -0.8\\ 0.0 & -0.6 & -1.3 & -1.8 & 1.9 \end{array}\right)\Rightarrow W_{\mathrm{SA}}=\left(\begin{array}{ccccc} 0.1 & 4.5 & -1.2 & 0.7 & 0.9\\ -0.7 & -2.6 & 1.8 & 0.0 & 0.5\\ 0.0 & -0.3 & -3.8 & 0.6 & 0.6\\ 0.5 & -1.1 & -3.4 & -1.0 & 0.7 \end{array}\right)\\ \; & \;B_{\mathrm{LQF}}={\left(0.5,-0.1,-0.2,-0.2\right)}^{\prime }\;\;\;\Rightarrow B_{\mathrm{SA}}={\left(-2.0,-0.3,0.3,-1.0\right)}^{\prime }. \end{array}$$



The average costs for heuristic policies take the values $g_{\mathrm{LQF}}=3.7333$, $g_{c\mu }=2.8000$, $g_{\mathrm{PIA}}=1.6500$ and $g_{\mathrm{LQF}}=5.0133$, $g_{c\mu }=3.9866$, $g_{\mathrm{PIA}}=2.7373$.

It is observed that the optimal policy obtained by the SA algorithm is quite close to those obtained by the PIA. Nevertheless, from experiment to experiment certain deviations in the value of the average costs may appear. Therefore it is of interest for us to check whether such differences are statistically significant.

Further we analyse how sensitive is the optimal policy obtained in exponential case by the SA algorithm to the shape of arrival and service time distributions. The following distributions will be used to calculate the optimal control policy in the non-exponential case: gamma G(α,β), log-normal LN(μ,σ) and Pareto PR(α,k) distributions, where two last options belong to a set of heavy tail distributions. The parameters of these distributions are chosen so that their first and second moments coincide. Moreover, the first moments are the same as for exponential distributions. The moments need to be represented as functions depending on the corresponding sample moments as in the method of moments used for parameter estimation. In the following experiments, the first moments of the inter-arrival and service times are fixed at values of Case 2, and the squared coefficient of variation is varied as $CV_{\nu_{i}}^{2}=CV_{\zeta_{i}}^{2}=0.5$ and $CV_{\nu_{i}}^{2}=CV_{\zeta_{i}}^{2}=20$. Denote by ${\left\{Z^{\left(k\right)}\right\}}_{k=1}^{m}$ a sample random variable *Z* distributed according to the proposed distributions with two first sample moments $\bar{Z}$, ${\bar{Z}}_{2}$ and squared empirical coefficient of variation $CV_{Z}^{2}=\frac{{\bar{Z}}_{2}}{\bar{Z}}-1$. Then for the gamma distribution Z∼G(α,β) with a PDF
$$f_{Z}\left(z\right)=\left\{\begin{array}{cc} \frac{\beta {\left(\beta z\right)}^{\alpha -1}e^{-\beta z}}{\Gamma (\alpha )} & z\geq 0,\\ 0 & z<0 \end{array}\right.$$
the parameters α>0 and β>0 satisfy the relations,
$$\alpha =\frac{1}{CV_{Z}^{2}},\;\beta =\frac{\alpha }{\bar{Z}}.$$

In case of the lognormal distribution Z∼LN(μ,σ) with a PDF
$$f_{Z}\left(z\right)=\frac{1}{\sigma z}\Phi \left(\frac{\ln(z)-\mu }{\sigma }\right),\;z>0,$$
the parameters μ∈R and σ>0 are calculated by
$$\sigma =\sqrt{\ln(1+CV_{Z}^{2})},\;\mu =\ln\left(\bar{Z}\right)-\frac{\sigma^{2}}{2}.$$

In case of a Pareto distribution Z∼PR(k,α) with a PDF
$$f_{Z}\left(z\right)=\left\{\begin{array}{cc} \frac{\alpha k^{\alpha }}{x^{\alpha +1}} & x\geq k\\ 0 & x<k \end{array}\right.$$
the parameters k>0 and α>0 are calculated by relations
$$\alpha =1+\frac{\sqrt{1+CV_{Z}^{2}}}{CV_{Z}},\;k=\frac{\alpha -1}{\alpha }\bar{Z}.$$
Parameters of the proposed probability distributions are listed in Table 5 and Table 6, respectively, for inter-arrival and service time distributions.

The sensitivity of the optimal control policy to the shape of the distributions is tested by means of a two-sided *t*-test for samples with unknown but equal variances. Let $g_{\exp}$ and $g_{\mathrm{opt}}$ are the samples of the average cost values obtained for the optimal control policy in case of exponentially distributed times and for the system with proposed distributions for the inter-arrival and service times. These samples of size *m* are associated with the normally distributed random variables $Z_{\exp}∼N\left(\mu_{g_{\exp}},\sigma_{g_{\exp}}\right)$ and $Z_{\mathrm{opt}}∼N\left(\mu_{g_{\mathrm{opt}}},\sigma_{g_{\mathrm{opt}}}\right)$, where $\mu_{g_{\exp}},\mu_{g_{\mathrm{opt}}}\in R$ and $\sigma_{g_{\exp}}=\sigma_{g_{\mathrm{opt}}}>0$. The test is defined then as
$$\begin{array}{c} H_{0}:\mu_{g_{\exp}}=\mu_{g_{\mathrm{opt}}}\;H_{1}:\mu_{g_{\exp}}\neq \mu_{g_{\mathrm{opt}}}\;p=P\left[\frac{|{\bar{g}}_{\exp}-{\bar{g}}_{\mathrm{opt}}|}{S_{g_{\mathrm{opt}},g_{\exp}}}>t_{2m-2;1-\frac{\alpha }{2}}\right], \end{array}$$
where statistics $S_{g_{\mathrm{opt}},g_{\exp}}$ is calculated by (Equation 16). The results of tests in form of the *p*-value, the values of the average costs ${\bar{g}}_{\exp}$ and ${\bar{g}}_{\mathrm{opt}}$ together with their 95% confidence intervals are summarized in Table 7 and Table 8 for the systems with different inter-arrival and service time distributions with smaller and greater levels of dispersion around the mean, d.h. for $CV_{\nu_{i}}^{2}=CV_{\zeta_{i}}^{2}=0.5$ in Table 7 and $CV_{\nu_{i}}^{2}=CV_{\zeta_{i}}^{2}=20$ in Table 8. Table cell contains two rows with the values for the average costs ${\bar{g}}_{\exp}$ and ${\bar{g}}_{\mathrm{opt}}$ together with confidence boundaries, and the third row has the *p*-value.

From the numerical examples, it is observed that the shape of distributions expressed through a coefficient of variation has a high level of influence over the value of the average cost functions ${\bar{g}}_{\exp}$ and ${\bar{g}}_{\mathrm{opt}}$. In almost all cases, the average cost increases significantly when the coefficient of variation increases. Only in the case of the Pareto distribution for the inter-arrival and service times is the change in values not significant. However, an examination of the entries in the last two tables reveals that in all experiments the *p*-value exceeds the significance level of α=0.05. Furthermore, it is worth noting that in most cases this exceeding is sufficient large. In this regard, the statistical test fails to reject null hypothesis at a given significance level, in other words, the average cost values are statistically equal and the corresponding optimal control policies are equivalent. Therefore, at least within the framework of the experiments conducted, we can state that the optimal scheduling policy is insensitive to the shape of the inter-arrival and service time distributions given that the first moments are equal. For practical purposes, in general queueing systems one can either apply the proposed optimization method, or use the control policy optimized for the equivalent exponential model as a suboptimal scheduling policy.

## 8. Conclusions

In this paper, we combined the queue simulation technique, neural network and simulated annealing optimization to calculate the optimal scheduling policy and optimized average cost function in a general single-server queueing system with multiple parallel queues. The proposed combination of tools is sufficiently versatile to solve discrete optimization problems that occur during resource allocation in complex queueing systems and networks. The numerical results subsequently demonstrate the effectiveness of the proposed approach. The obtained optimal scheduling policy outperforms the best available heuristic policy which is the cμ-rule by more than 45% on average. Nevertheless, a couple of important points must be stressed that can be considered when using the proposed method. In simulated annealing, the choice of initial control policy affects the speed of convergence to the optimal solution. Furthermore, it is required that the finite domain be defined for the solution. If the dimensionality of the state space allows, the initial control policy and the corresponding finite solution space can be obtained by the policy iteration algorithm implemented for the Markov model. The obtained optimal solution seems to be statistically insensitive to the form of inter-arrival and service time distributions where the first two moments are the same. Moreover, the optimal policy in exponential case can be treated as a suboptimal policy and the corresponding trained neural network can be used by routers in queueing systems with arbitrary distributions. In terms of future research, we see potential in developing and applying this method to other complex controlled queueing systems where the search for optimal routing, scheduling and resource allocation policies is required. The possibility to compose the reinforcement learning algorithms and neural networks to solve optimization problems in general controlled queueing models could also be considered as a further line of research.

## Figures and Tables

**Figure 1 sensors-23-05479-f001:**
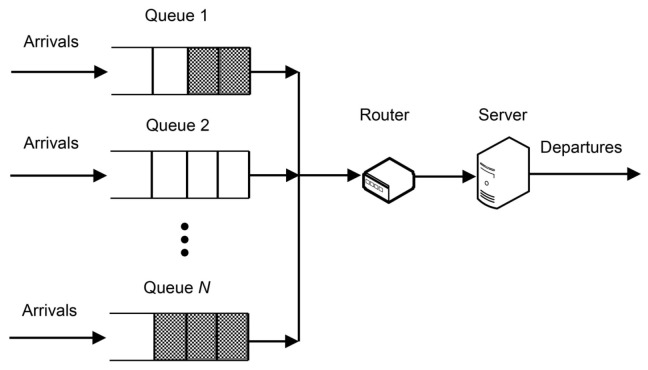
Controlled single-server queueing system with parallel queues.

**Figure 2 sensors-23-05479-f002:**
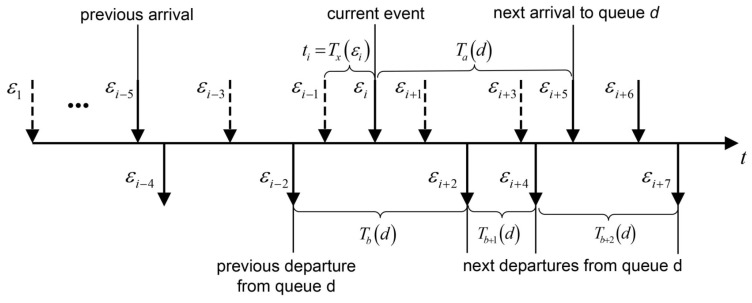
The time assignment for the present time based simulation.

**Figure 3 sensors-23-05479-f003:**
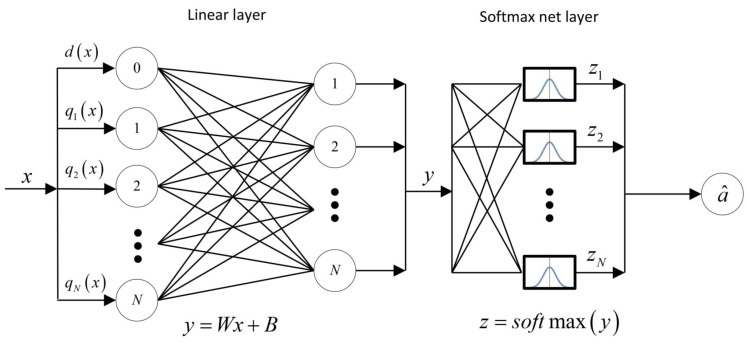
Neural network architecture.

**Figure 4 sensors-23-05479-f004:**
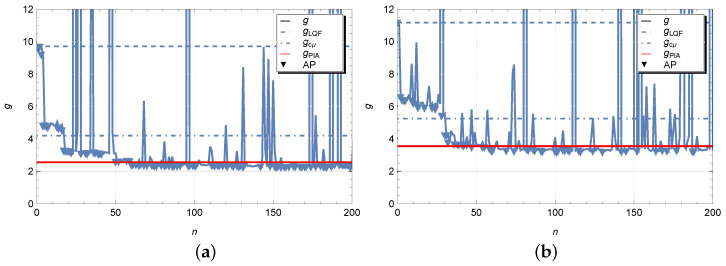
Iteration steps for *g* with $\nu_{i}∼E\left(\lambda_{i}\right)$ and $\zeta_{i}∼E\left(\mu_{i}\right)$ for Case 1 (**a**) and Case 2 (**b**).

**Figure 5 sensors-23-05479-f005:**
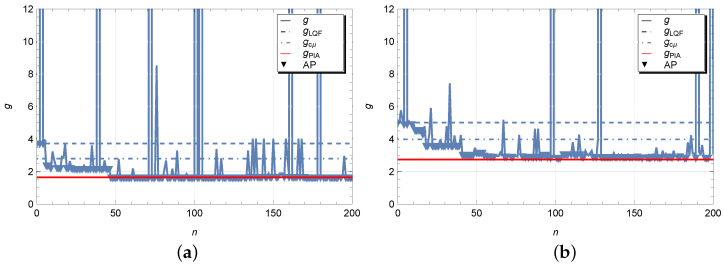
Iteration steps for *g* with $\nu_{i}=\frac{1}{\lambda_{i}}$ and $\zeta_{i}=\frac{1}{\mu_{i}}$ for Case 1 (**a**) and Case 2 (**b**).

**Table 1 sensors-23-05479-t001:** The values of system parameters.

*i*	1	2	3	4
$\lambda_{i}$	0.05	0.10	0.15	0.20
$\mu_{i}$	3.750	1.875	1.250	0.938
$c_{i}$	1	1	1	1
$c_{i,1}$	0	1	2	3
$c_{i,2}$	3	0	1	2
$c_{i,3}$	2	3	0	1
$c_{i,4}$	1	2	3	0

**Table 2 sensors-23-05479-t002:** The optimal scheduling policy for selected states.

$\left(d,q_{1},q_{2},q_{3},q_{4}\right)$	0	1	2	3	4	5	6	7	8	9	10
$\left(1,0,q_{2},1,1\right)$	4	4	4	4	4	4	4	4	4	4	4
$\left(1,0,q_{2},3,3\right)$	4	4	4	4	4	3	3	3	3	3	3
$\left(1,0,q_{2},5,5\right)$	4	3	3	3	3	3	3	2	2	2	2
$\left(1,0,q_{2},8,8\right)$	3	3	3	3	2	2	2	2	2	2	2
$\left(1,0,q_{2},9,9\right)$	3	3	2	2	2	2	2	2	2	2	2
$\left(2,q_{1},0,1,1\right)$	4	4	4	4	4	4	4	4	4	4	4
$\left(2,q_{1},0,3,3\right)$	4	4	4	4	4	4	3	3	3	3	3
$\left(2,q_{1},0,5,5\right)$	3	3	3	3	3	3	3	3	3	3	3
$\left(2,q_{1},0,8,8\right)$	3	3	3	3	3	3	3	3	3	3	3
$\left(2,q_{1},0,9,9\right)$	3	3	3	3	3	3	3	3	1	1	1

**Table 3 sensors-23-05479-t003:** The values of arrival rates.

*i*	1	2	3	4
$\lambda_{i}$	0.08	0.16	0.24	0.32

**Table 4 sensors-23-05479-t004:** The optimal scheduling policy for selected states.

$\left(d,q_{1},q_{2},q_{3},q_{4}\right)$	0	1	2	3	4	5	6	7	8	9	10
$\left(1,0,q_{2},1,1\right)$	3	2	2	2	2	2	2	2	2	2	2
$\left(1,0,q_{2},3,3\right)$	3	2	2	2	2	2	2	2	2	2	2
$\left(1,0,q_{2},5,5\right)$	3	2	2	2	2	2	2	2	2	2	2
$\left(1,0,q_{2},8,8\right)$	3	2	2	2	2	2	2	2	2	2	2
$\left(1,0,q_{2},9,9\right)$	3	2	2	2	2	2	2	2	2	2	2
$\left(2,q_{1},0,1,1\right)$	3	3	1	1	1	1	1	1	1	1	1
$\left(2,q_{1},0,3,3\right)$	3	3	1	1	1	1	1	1	1	1	1
$\left(2,q_{1},0,5,5\right)$	3	1	1	1	1	1	1	1	1	1	1
$\left(2,q_{1},0,8,8\right)$	3	1	1	1	1	1	1	1	1	1	1
$\left(2,q_{1},0,9,9\right)$	3	1	1	1	1	1	1	1	1	1	1

**Table 5 sensors-23-05479-t005:** Parameters for inter-arrival time distributions, $CV_{\nu_{i}}^{2}=0.5$ (**a**) and $CV_{\nu_{i}}^{2}=20$ (**b**).

(a)				
i	**1**	**2**	**3**	**4**
$G(\alpha_{i},\beta_{i})$	(2.00,0.16)	(2.00,0.32)	(2.00,0.48)	(2.00,0.64)
$\mathrm{LN}(m_{i},\sigma_{i})$	(2.323,0.637)	(1.629,0.637)	(0.937,0.637)	(0.637,0.637)
$\mathrm{PR}(k_{i},\alpha_{i})$	(7.925,2.732)	(3.962,2.732)	(2.642,2.732)	(1.981,2.732)
**(b)**				
i	**1**	**2**	**3**	**4**
$G(\alpha_{i},\beta_{i})$	(0.05,0.004)	(0.05,0.008)	(0.05,0.012)	(0.05,0.016)
$\mathrm{LN}(m_{i},\sigma_{i})$	(1.003,1.745)	(0.310,1.745)	(−0.095,1.745)	(−0.383,1.745)
$\mathrm{PR}(k_{i},\alpha_{i})$	(6.326,2.025)	(3.163,2.025)	(2.109,2.025)	(1.582,2.025)

**Table 6 sensors-23-05479-t006:** Parameters for service time distributions, $CV_{\zeta_{i}}^{2}=0.5$ (**a**) and $CV_{\zeta_{i}}^{2}=20$ (**b**).

(a)				
i	**1**	**2**	**3**	**4**
$G(\alpha_{i},\beta_{i})$	(2.00,7.500)	(2.00,3.750)	(2.00,2.500)	(2.00,1.875)
$\mathrm{LN}(m_{i},\sigma_{i})$	(−1.524,0.637)	(−0.831,0.637)	(−0.426,0.637)	(−0.138,0.637)
$\mathrm{PR}(k_{i},\alpha_{i})$	(0.169,2.732)	(0.338,2.732)	(0.507,2.732)	(0.676,2.732)
**(b)**				
i	**1**	**2**	**3**	**4**
$G(\alpha_{i},\beta_{i})$	(0.05,0.198)	(0.05,0.094)	(0.05,0.063)	(0.05,0.047)
$\mathrm{LN}(m_{i},\sigma_{i})$	(−2.844,1.745)	(−2.151,1.745)	(−1.745,1.745)	(−1.458,1.745)
$\mathrm{PR}(k_{i},\alpha_{i})$	(0.135,2.025)	(0.269,2.025)	(0.405,2.025)	(0.539,2.025)

**Table 7 sensors-23-05479-t007:** Comparison of optimal policies for $CV_{\nu_{i}}^{2}=CV_{\zeta_{i}}^{2}=0.5$.

	Arrival	G	LN	PR
Service				
G		3.0836±0.0286 3.0491±0.0576 p=0.2964	3.0556±0.0196 3.0203±0.0301 p=0.0569	3.0096±0.0437 3.0083±0.0726 p=0.9736
LN		3.0818±0.0291 3.0445±0.0233 p=0.0527	3.0654±0.0227 3.0282±0.0364 p=0.0931	3.0347±0.0622 3.0351±0.0877 p=0.9881
PR		3.0904±0.0485 3.0168±0.0701 p=0.0942	3.1142±0.0539 3.0572±0.0614 p=0.1749	3.3081±0.4249 3.1435±0.1305 p=0.4709

**Table 8 sensors-23-05479-t008:** Comparison of optimal policies for $CV_{\nu_{i}}^{2}=CV_{\zeta_{i}}^{2}=20$.

	Arrival	G	LN	PR
Service				
G		44.5518±5.3662 40.8015±4.0916 p=0.2788	48.3524±13.0935 38.6532±15.3943 p=0.3493	19.1573±3.7810 16.3102±1.6154 p=0.1793
LN		44.0659±4.6092 41.6925±5.4512 p=0.5162	26.7610±6.0684 28.8180±8.7892 p=0.7067	9.6126±1.9352 11.9165±4.6811 p=0.3759
PR		36.3436±4.0311 34.1937±2.4608 p=0.3749	32.9247±11.1232 24.4347±4.1215 p=0.1656	5.6667±0.7101 6.4067±1.5618 p=0.4008

## Data Availability

The authors can be contacted to obtain data used in the study.

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
