# Peer review of "Optimal Scheduling in General Multi-Queue System by Combining Simulation and Neural Network Techniques"

_sensors, 2023, doi:10.3390/s23125479_

Round 1

Reviewer 1 Report

The paper’s scope is within the scope of the journal, and it presents an original contribution. The abstract is somehow sufficient to give useful information about the paper’s topic. The proposed algorithm is somehow described and thoroughly illustrated. The paper is somehow well-structured and written, and the text is not clear and easy to read. However, there are some comments we recommend the authors to do:

1. The title is too long; please shorten it as required.

2.  Abstract is not able to convey what is the technical contribution of this paper. I suggest to re-write it.

3. Introduction is too much. Explain more relevant concepts and up-to-date work.

4. In the introduction section cited references are old. Please cite recent papers relevant to the concepts. 

5. The paper should be also supported by a literature search including relevant and recent papers. The following articles related to NN may be cited both.

* https://doi.org/10.3390/tropicalmed7120424

* https://doi.org/10.1111/exsy.13105 

6. At the end of the conclusions section (Section 8), present your best results in terms of various performance metrics as values or percentages. 

- I recommend the authors thoroughly proofread the manuscript to correct all the typos.

The paper is very interesting, but it is affected by poor English text. Before its eventual acceptance, very extensive English editing is forcibly required.

Author Response

We would like to thank the reviewer very much for his/her very helpful comments and remarks. Our comments on each suggestion are listed below.

  1. The title is too long; please shorten it as required..

The title of the article is made shorter.

  1. Abstract is not able to convey what is the technical contribution of this paper. I suggest to rewrite it.

We have rewritten the abstract and highlighted the contribution of this article.

  1. Introduction is too much. Explain more relevant concepts and up-to-date work.

The application of multilevel neural networks in queueing systems is a rather new topic. For the purpose of this paper we want to give a small overview of existing publications, so the introduction is a bit longer than usual. Since there are various sections related to managed mass-service systems, optimization, etc., it was necessary to give some references for each such section.  We have restructured the introduction and the individual sections now have a more logical connection.

  1. In the introduction section cited references are old. Please cite recent papers relevant to the concepts..

The introduction begins with a brief overview of articles related to the use of machine learning, in particular neural networks, in the models of queueing theory. If one looks carefully, these papers are new, published between 2017 and 2022. However, we have added another of the more recent papers from 2022. The references to older works refer to the basic classical methods, which are modified accordingly in this paper.

  1. The paper should be also supported by a literature search including relevant and recent papers. The following articles related to NN may be cited both.

Thank you so much for the links to such interesting works. We realise that there is a huge amount of papers related to the use of neural networks in various fields of science and technics. However, in our work, we do not focus on neural networks themselves, but rather on their connection to controlled queueing systems. The references given in the paper follow this logic.

  1. At the end of the conclusions section (Section 8), present your best results in terms of various performance metrics as values or percentages.

We have added a relevant result in the conclusion.

  1. I recommend the authors thoroughly proofread the manuscript to correct all the typos.

We have made changes to the text, changed the structure of a number of sentences and corrected typos. If the reviewer has any comments and remarks on the new version, please indicate the specific place for corrections.

Reviewer 2 Report

The research paper has some interesting  findings, which may significantly contribute into the literature. However, there are some minor issues:
1. An abstract should address these questions: what are you trying to do, why, what you found and what is the significance of your findings. Rewrite and improve.
2. There are several  models already developed in the literature. What is the utility of your proposed model.
3. Comparative discussion needs further explanation. Please do more work on it.
4. Please improve the conclusion section. Also, limitations in the developed approach should be discussed in the conclusions section.
5. The notations used should be rechecked.
6. The writing is recommended to be improved. The authors are suggested to proofread paper and restructuring of sentences are required for the entire manuscript.

To be improved.

Author Response

We would like to thank the reviewer very much for his/her very helpful comments and observations. We have tried to make the following corrections. Below there are our comments on each of the reviewer's comments.

  1. An abstract should address these questions: what are you trying to do, why, what you found and what is the significance of your findings. Rewrite and improve.

We have rewritten the abstract, where we have clearly highlighted what the work explores specifically and why, as well as what significant result has been obtained.

  1. There are several models already developed in the literature. What is the utility of your proposed model.

The main idea of the paper is to use a multilayer neural network (NN) for server scheduling. The parameters of this NN are optimized with the aim to minimize a given average cost function. Moreover, such a cost function for systems with arbitrary inter-arrival and service time distributions can only be computed by a simulation. We consider this approach, which combines neural networks with simulation technique, to be quite universal and illustrate it by the example of the queueing system described in the paper. Such a system with arbitrary distributions and switching costs has not yet been considered by other authors.

  1. Comparative discussion needs further explanation. Please do more work on it.

It is not quite clear which part of the comparative analysis should be further revised but we have added a number of explanations to better understand the results presented. If the reviewer has specific suggestions for improvement we will certainly follow up on the recommendations.

  1. Please improve the conclusion section. Also, limitations in the developed approach should be discussed in the conclusions section.

The only limitation of this approach concerns the choice of initial control policy in the simulated annealing algorithm. The worse the initial policy is in terms of the cost function, the more iterative cycles need to be applied to obtain an adequate optimal solution. This remark has been included in the conclusion.

  1. The notations used should be rechecked.

We found a few places with typos in notations. If there are inaccuracies in the revised version, we kindly ask the Reviewer to indicate the specific location.

  1. The writing is recommended to be improved. The authors are suggested to proofread paper and restructuring of sentences are required for the entire manuscript.

We did some additional editing work. A number of sentences and phrases have been rewritten, and typos have also been corrected.

Reviewer 3 Report

In this study, the authors explore the possibility of combining simulation and neural network paradigms for optimal scheduling in a heterogeneous system with an arbitrary inter-arrival and service time distributions. The idea is interesting. Some comments are as follows:

1. The motivation is not clear.

2. The abstract needs to be written a bit more to highlight its own contribution.

3. What is the core theoretical contribution of the paper?

4. It is suggested to add a paragraph to explain the innovation of this work.

5. In terms of literature research, it is suggested to add the description for the following machine learning work:  A fuzzy cluster validity index induced by triple center relation. IEEE Transactions on Cybernetics.

6. The paper needs to say more about a vision of future work.

7. Table 3: Why does the author use these parameters?

The quality of English needs to be improved.

Author Response

Response to Reviewer 3.

We would like to thank the reviewer very much for his/her very helpful comments and remarks. Our comments on each suggestion are listed below.

  1. The motivation is not clear.

We have made changes to the abstract and to the introduction, where we have explained the motivation more explicit. The basic idea is that finding optimal scheduling policy in queueing systems with arbitrary distributions is a non-trivial problem. We have proposed to use a neural network as the source of information for the controller. This network is trained on some arbitrary or heuristic control policy, and then the parameters of this trained network are optimized with the aim to minimize the loss functional.

  1. The abstract needs to be written a bit more to highlight its own contribution.

We have rewritten the abstract and highlighted the contribution of this article.

  1. What is the core theoretical contribution of the paper?

The key theoretical result is to prove the effectiveness of using a neural network for scheduling problem in queueing systems with arbitrary inter-arrival and service time distributions.  It is shown that the parameters of such a network in the form of weights and biases can be optimized in such a way so that a new neural network minimizes the average cost, which in turn can only be calculated by simulation. For the optimization algorithm, it was necessary to determine the initial values of the neural network parameters, the domain of these parameters, which would cover the optimal solution. As suggested in the paper, such an area can be estimated using the parameters of the neural network which is trained on the data obtained for the corresponding Markov model.

  1. It is suggested to add a paragraph to explain the innovation of this work.

At the end of introduction on page 4, we summarise the contribution made in this paper.

  1. In terms of literature research, it is suggested to add the description for the following machine learning work...

Thank you very much for the links to such interesting works. However, in this paper we wanted to focus on combining neural network and simulation techniques specifically in the theory of managed mass service systems, and all the references proposed are in one way or another related to this theory.

  1. The paper needs to say more about a vision of future work.

Our vision of the potential for further development of this topic was added to the conclusion section.

  1. Table 3: Why does the author use these parameters?

The parameter values are chosen to reflect the heterogeneity of the parallel queues. Holding costs in the queue for simplicity are assumed to be equal to 1. Thank you for bringing this table to your attention. We have found typos in the values of the sw

Reviewer 4 Report

In my opinion, the paper is interesting and well-written. The topic is relevant and addresses some of the key technologies (AI, for example) nowadays.

The experiments are correct and results support the conclusions and the hypotheses, although some improvements are needed. The technological descriptions are exhaustive and the proposal sound.

In my opinion, the paper may be accepted if some changes are implemented:

- First, I think an "state of the art" section needs to included. Many references should be discussed and analyzed. You need to clearly describe the open problems and questions in the area. And explain how your proposal solves some of them. And why this new approach is different from the previous 

- Second, in your experimental section some comparisons between the current solutions and your approach must be provided. Statistical tests need to be performed before concluding your solution improves the state of the art

Author Response

Response to Reviewer 4.

We would like to thank the reviewer very much for his/her very helpful comments and remarks. Our comments on each suggestion are listed below.

  1. I think an "state of the art" section needs to included. Many references should be discussed and analysed. You need to clearly describe the open problems and questions in the area. And explain how your proposal solves some of them. And why this new approach is different from the previous.

We have rewritten the abstract and made structural changes to the introduction. The state of art is now adequately described in the introduction. The main objective of the article and the methodology to achieve it are described more clearly. The main contributions of the article are presented in a separate paragraph.

  1. In your experimental section some comparisons between the current solutions and your approach must be provided. Statistical tests need to be performed before concluding your solution improves the state of the art.

The conclusions proposed in the paper based on numerical experiments are supported by statistical hypothesis tests. The effectiveness of finding an optimal policy is demonstrated by several examples.

Reviewer 5 Report

General comments 

This study combined the queue simulation technique, neural network and simulated annealing optimization to calculate the optimal scheduling policy and optimized average cost function in a general single-server queueing system with multiple parallel queues. The proposed combination of tools is sufficiently versatile to solve discrete optimization problems occurring by resource allocation in complex queueing systems and networks.

In my view, the topic has the originality to be considered for publication as it covers the gap in the literature.

The authors provided a comprehensive literature review although some articles should be added. The findings indicate that the optimal policy in exponential case can be treated as a suboptimal policy and the corresponding trained neural network can be used by router in queueing systems with arbitrary distributions.

The conclusion has written in a proper format.

Some missing references should be added (please see my comments above).

Tables and figures are legible.

Detailed comments

The manuscript entitled “Optimal scheduling in a general single-server system with heterogeneous queues and switching costs using simulation and neural network paradigms” seems to be acceptable to be considered for publication in Sensors, but it would be better if some issues be regarded in the revision.
1. Since this paper is not a literature review, in lieu of referring to a vast number of previous works, which have been surveyed by other researchers, mention the latest related works.

Aljafari, Belqasem, et al. "Electric vehicle optimum charging-discharging scheduling with dynamic pricing employing multi agent deep neural network." Computers and Electrical Engineering 105 (2023): 108555.

Nobari, Arash, et al. "Developing a bi-objective model for a reliable mobile ad hoc network routing problem." Social Network Analysis and Mining 6 (2016): 1-6.

Zhan, Weipeng, et al. "A review of siting, sizing, optimal scheduling, and cost-benefit analysis for battery swapping stations." Energy (2022): 124723.

2. The transitions from topic to topic in the paper seem to be a little sudden. In other words, while reading about a topic, the text suddenly starts to mention something quite different. It is suggested to smooth these transitions from topic to topic where possible.
3. Please indicate the contributions in more detail, specifically in comparison with the latest research papers.
4. Above all, please polish the title to be in line with the contents of this manuscript.

It requires proofreading but it has the fluency in its current form.

Author Response

Response to Reviewer 5.

We would like to thank the reviewer very much for his/her very helpful comments and remarks. Our comments on each suggestion are listed below.

  1. Since this paper is not a literature review, in lieu of referring to a vast number of previous works, which have been surveyed by other researchers, mention the latest related works.

Many thanks for the links to very interesting papers. We have included one of these researches that fits well with this work.

  1. The transitions from topic to topic in the paper seem to be a little sudden. In other words, while reading about a topic, the text suddenly starts to mention something quite different. It is suggested to smooth these transitions from topic to topic where possible.

We changed the structure of the introduction, explaining individual elements more clearly. The introduction now has a more logical connection between the individual paragraphs.

  1. Please indicate the contributions in more detail, specifically in comparison with the latest research papers.

Significant contributions were highlighted in a separate paragraph.

  1. Above all, please polish the title to be in line with the contents of this manuscript.

We have rewritten the title of the paper.

Round 2

Reviewer 2 Report

After my careful observation of this revised version, the present form of the paper can be accepted for publication.

After my careful observation of this revised version, the present form of the paper can be accepted for publication.

Reviewer 4 Report

In my opinion, the authors have addressed all my previous concerns and the paper may be accepted.